

# Estimation of common percentile of rainfall datasets in Thailand using delta-lognormal distributions

Warisa Thangjai[1], Sa-Aat Niwitpong[2] and Suparat Niwitpong[2]

[1] Department of Statistics, Faculty of Science, Ramkhamhaeng University, Bangkok, Thailand
[2] Department of Applied Statistics, Faculty of Applied Science, King Mongkut's University of Technology North Bangkok, Bangkok, Thailand

## ABSTRACT

Weighted percentiles in many areas can be used to investigate the overall trend in a particular context. In this article, the confidence intervals for the common percentile are constructed to estimate rainfall in Thailand. The confidence interval for the common percentile help to indicate intensity of rainfall. Herein, four new approaches for estimating confidence intervals for the common percentile of several delta-lognormal distributions are presented: the fiducial generalized confidence interval, the adjusted method of variance estimates recovery, and two Bayesian approaches using fiducial quantity and approximate fiducial distribution. The Monte Carlo simulation was used to evaluate the coverage probabilities and average lengths *via* the R statistical program. The proposed confidence intervals are compared in terms of their coverage probabilities and average lengths, and the results of a comparative study based on these metrics indicate that one of the Bayesian confidence intervals is better than the others. The efficacies of the approaches are also illustrated by applying them to daily rainfall datasets from various regions in Thailand.

## INTRODUCTION

Water is necessary for humans and the planet's ecosystem. The amount of water that is available usually depends on the amount of rain that falls. A drought occurs when an abnormally long dry period uses up available water resources. Floods occur when rain swallows up land that is usually uncovered. These natural disasters are often made worse by human action. Thailand is divided into six geographical regions. Precipitation is not distributed equally around the country. Some regions barely see rain while others get more than their share. For example, heavy rains in central and northeastern Thailand caused flooding in 2021. The droughts and floods are a common feature and poses natural disaster. The droughts and floods cannot be eradicated but has to be managed. As a result, the common percentile is more appropriate than the common variance or the common coefficient of variation. Therefore, the common percentile is used to describe the rainfall dispersion in different regions.

Corresponding author
Sa-Aat Niwitpong,
sa-aat.n@sci.kmutnb.ac.th

Distributions of life science data are often skewed and contain a relatively large number of zero observations. In some situations, it is appropriate to model such data using a log-normal distribution for the positive values together with the probability of the point mass at zero. This distribution, commonly referred to as delta-lognormal (*Aitchison, 1955*), is often used to analyze environmental and medical data. For example, *Zhou & Tu (2000)* presented interval estimation for the mean of diagnostic test charge data for older adults with depression. Rainfall amount data series often contain zero values, which makes the delta-lognormal distribution appropriate for fitting them. *Maneerat, Niwitpong & Niwitpong (2020)* and *Maneerat & Niwitpong (2021)* proposed interval estimation for daily rainfall amount data in Thailand. *Yosboonruang & Niwitpong (2021)* studied confidence intervals for the ratio of two independent coefficients of variation of delta-lognormal distributions and demonstrated their efficacies with rainfall amount data from Thailand.

In some cases, the tails of several distributions can be different even though their means are the same. The percentile is a measure indicating the value below which a given percentage of observations in a group falls and can be used to describe key aspects of a distribution such as the central tendency and spread. It can play a significant role in the day-to-day statistical analyses of data and, when used in certain situations, can be more useful than the mean because it can be used to interpret a specific item, individual, or unit, which is not possible with the mean. The 25th and the 75th percentiles are the first and the third quartiles, respectively, and the interquartile range (the difference between the third and the first quartiles) is often used as a measure of spread in a distribution. Furthermore, the most well-known (50th) percentile is the median, which has been applied in statistic inference for diverse areas such as household income and survival time. The 95th percentile has been used to set premiums in the insurance industry by conducting interval estimation for normal distributions (*Chakraborti & Li, 2007*) and to compute the event rainfall depths (*Shrestha, Fang & Zech, 2014*). *Krishnamoorthy, Mathew & Xu (2013)* presented hypothesis testing for the upper percentile of a log-normal distribution based on samples with multiple detection limits and sample-size calculations.

Extreme rainfall events can lead to flooding. The extreme rainfall events play an important role in disaster prevention and preparedness. In practice, the extreme rainfall is practically quantified by estimating the extreme rainfall quantiles *via* quantiles estimation of various hydrological variables through selected statistical models using measured data. However, the measured data contain inherent randomness and uncertainty, the estimated quantiles are also uncertain. Therefore, the confidence intervals are used to estimate the rainfall information for reducing uncertainty and making hydrologic analysis effective (*Chen et al., 2016*; *Serinaldi, 2009*). Several researchers have studied the percentile or quantile of the rainfall data. For instance, *Chen et al. (2016)* constructed the confidence intervals of extreme rainfall quantiles using Bayesian, bootstrap, and profile likelihood approaches. *Serinaldi (2009)* constructed the confidence intervals for quantiles with emphasis on the extreme ones. *Reis & Stedinger (2005)* compared the confidence intervals for quantiles with historical information using the maximum likelihood estimator and Bayesian approaches. *Lu et al. (2013)* presented the confidence intervals for quantiles of extreme value distribution of hydrometeorology.

Let $k$ be number of sample cases. For $k \geq 2$ independent samples of independent identically distributed random variables both with the same parameter ($\theta$), one can pool the samples from different sources for inference purposes based on this common parameter. Herein, we are concerned with estimating common parameter $\theta$. The problem of constructing confidence intervals for the common parameter from multiple distributions has been solved by several researchers. For example, *Krishnamoorthy & Lu (2003)* proposed the generalized variable approach for inference on the common mean of normal distributions. *Lin & Lee (2005)* developed a new generalized pivotal quantity based on the best linear unbiased estimator for constructing confidence intervals for the common mean of normal distributions. *Tian (2005)* presented procedures for inference on the common coefficient of variation of normal distributions. *Tian & Wu (2007)* provided the generalized variable approach for inference on the common mean of log-normal distributions. *Ye, Ma & Wang (2010)* presented procedures for hypothesis testing and interval estimation for the common mean of inverse Gaussian distributions. *Ng (2014)* constructed a confidence interval for the common coefficient of variation of several independent log-normal samples based on the generalized confidence interval (GCI) approach. *Thangjai, Niwitpong & Niwitpong (2017a)* investigated a new confidence interval for the common mean of several normal distributions by using an adjusted method of variance estimates recovery (Adjusted MOVER) approach. *Thangjai & Niwitpong (2017)* provided novel approaches to construct confidence intervals for the common coefficient of variation of weighted two-parameter exponential distributions. *Thangjai, Niwitpong & Niwitpong (2017b)* studied confidence intervals for the common mean of several one-parameter exponential populations. *Thangjai & Niwitpong (2018)* presented confidence intervals for common variance of several one-parameter exponential populations. *Thangjai, Niwitpong & Niwitpong (2020a)* proposed adjusted generalized confidence intervals for the common coefficient of variation of several normal populations. *Thangjai & Ruengpeerakul (2020)* constructed confidence intervals for the common inverse mean of several normal populations with unknown coefficients of variation. *Thangjai & Niwitpong (2020a)* established confidence intervals for the common signal-to-noise ratio of several log-normal distributions. *Thangjai, Niwitpong & Niwitpong (2020b)* presented confidence intervals for the common coefficient of variation of several rainfall data series in Thailand.

The problem of estimating parameters of interest from both classical and Bayesian perspectives has been extensively studied. For the classical methodology, the fiducial generalized confidence interval (FGCI) and adjusted MOVER approaches have been widely used to construct confidence intervals for the parameter of interest. For example, *Hannig et al. (2006)* introduced an FGCI approach to estimate simultaneous confidence intervals for the ratio of means of log-normal distributions. *Kharrati-Kopaei, Malekzadeh & Sadooghi-Alvandi (2013)* constructed simultaneous FGCIs for the successive differences of exponential location parameters under heteroscedasticity. *Zhang & Falk (2014)* proposed simultaneous confidence intervals constructed by using the FGCI approach for the ratio of the means of several log-normal distributions with heteroscedastic variances and unequal group sizes. *Thangjai, Niwitpong & Niwitpong (2016)* estimated simultaneous

confidence intervals for the differences between the coefficients of variation of several log-normal distributions based on the FGCI approach. The FGCI approach can be used to estimate the confidence interval for complex parameters, but this approach is based on the simulation data for constructing the confidence interval. *Thangjai & Niwitpong (2017)* proposed the adjusted MOVER approach to estimate the confidence interval for the weighted coefficients of variation of two-parameter exponential distributions. *Thangjai, Niwitpong & Niwitpong (2017a)* proposed the confidence intervals for the common mean of several normal populations based on the adjusted MOVER approach. *Thangjai, Niwitpong & Niwitpong (2017b)* proposed the adjusted MOVER approach for constructing the confidence intervals for the common mean of several one-parameter exponential populations. The adjusted MOVER approach is easy to construct the confidence interval using the exact formula, but this approach is based on the initial confidence interval for the single parameter.

For the Bayesian methodology, *Thangjai, Niwitpong & Niwitpong (2020b)* proposed confidence intervals for the common coefficient of variation of log-normal distributions based on a Bayesian approach. *Maneerat, Niwitpong & Niwitpong (2020)* constructed Bayesian credible intervals for the difference between variances of delta-lognormal distributions. *Yosboonruang, Niwitpong & Niwitpong (2020)* proposed Bayesian credible intervals for measuring the difference between rainfall dispersions datasets for regions in Thailand. *Thangjai, Niwitpong & Niwitpong (2020c)* estimated Bayesian credible intervals for the means of normal distributions with unknown coefficients of variation. *Thangjai, Niwitpong & Niwitpong (2021a)* applied Bayesian credible intervals for the coefficients of variation of PM10 dispersion datasets. *Thangjai, Niwitpong & Niwitpong (2021b)* applied the Bayesian approach for estimating the coefficients of variation of normal distributions. The Bayesian approach can be used to construct the confidence interval for complex parameters, but it is based on the prior distribution of parameter.

To the best of our knowledge, approaches based on interval estimation for the common percentile of delta-lognormal distributions have not previously been reported. The objective of this study, the confidence intervals using the FGCI, adjusted MOVER, and two Bayesian approaches for the common percentile of delta-lognormal distributions are constructed and their efficacies are compared. The confidence intervals were used to environmental model such as rainfall data. The common percentile can help to describe the 5th percentage of the heavy rainfall data in different regions.

## MATERIALS AND METHODS

For one population, let $V = (V_1, V_2, \ldots, V_n)$ be a positive random variable from a log-normal distribution with parameter $\mu$ and $\sigma^2$. The probability density function of $V_i$ is defined as

$$f(v_i; \mu, \sigma^2) = \begin{cases} \dfrac{1}{v_i \sigma \sqrt{2\pi}} \exp\left(-\dfrac{1}{2\sigma^2}(\ln(v_i) - \mu)^2\right); & v_i > 0 \\ 0; & otherwise \end{cases} \tag{1}$$

Delta-lognormal population contains both zero and non-zero observed values. Let $\delta' = 1 - \delta$ be the probability of zero observations and let $\delta$ be the probability of non-zero observations. Suppose that $n = n_{(0)} + n_{(1)}$ is the sample of size, where $n_{(0)}$ is a number of zeros and $n_{(1)}$ is a number of non-zeros. The zero observations follow a binomial distribution, $n_{(0)} \sim B(n, \delta')$. The non-zero observations follow a log-normal distribution. Let $X = (X_1, X_2, ..., X_n)$ be random sample drawn from a delta-lognormal distribution with parameters mean $\mu$, variance $\sigma^2$, and probability of obtaining a zero observation $\delta'$. Moreover, let $x = (x_1, x_2, ..., x_n)$ be the observed value of $X = (X_1, X_2, ..., X_n)$.

According to *Tian & Wu (2006)*, the probability density function of $X_i$ is given by

$$G(x_i; \mu, \sigma^2, \delta') = \begin{cases} \delta'; & x = 0 \\ \delta' + \delta F(x_i; \mu, \sigma^2); & x > 0 \end{cases}, \tag{2}$$

where $F(x_i; \mu, \sigma^2)$ is the log-normal cumulative distribution function.

Let $Y = (Y_1, Y_2, ..., Y_n)$ be independent log-transformed lognormal random variables. Let $\bar{Y}$ and $S^2$ be the mean and variance based on log-transformed sample from log-normal distribution. Also, $\bar{y}$ and $s^2$ be observed values of $\bar{Y}_{(1)}$ and $S^2$, respectively. Let $\bar{Y}_{(1)}$ and $S^2_{(1)}$ be the mean and variance based on the log-transformed positive sample measurements. And let $\bar{y}_{(1)}$ and $s^2_{(1)}$ be observed values of $\bar{Y}_{(1)}$ and $S^2_{(1)}$, respectively.

Following *Hasan & Krishnamoorthy (2018)*, let $q_p$ be the $p$th quantile of the delta-lognormal distribution. It is determined by the equation $G(q_p; \mu, \sigma^2, \delta') = p$. It follows Eq. (2) that $q_p = 0$ if $p < \delta'$. For $p > \delta'$, using the relation between the normal and log-normal distributions, the $q_p$ yields

$$q_p = \exp\left(\mu + \Phi^{-1}\left(\frac{p - \delta'}{1 - \delta'}\right)\sigma\right), \tag{3}$$

where $\phi$ is the standard normal distribution function.

Estimation of the $p$th quantile of the delta-lognormal distribution simplifies to estimation of

$$\lambda_p = \mu + \Phi^{-1}\left(\frac{p - \delta'}{1 - \delta'}\right)\sigma. \tag{4}$$

For $k$ populations, for $i = 1, 2, ..., k$, let $X_i = (X_{i1}, X_{i2}, ..., X_{in i})$ be a sample with $n_{i(0)}$ zeros from a delta-lognormal distribution. Let us assume that $n_{i(0)} \sim B(n_i, \delta'_i)$ so that $n_{i(1)} = n_i - n_{i(0)}$ sample measurements are greater than zero. Let $Y_i = \ln(X_i) \sim N(\mu_i, \sigma^2_i)$, where $X_i > 0$. Let $\bar{Y}_{i(1)}$ and $S^2_{i(1)}$ be the mean and variance based on the log-transformed positive sample measurements, respectively. Also, let $\bar{y}_{i(1)}$ and $s^2_{i(1)}$ be the observed values of $\bar{Y}_{i(1)}$ and $S^2_{i(1)}$, respectively.

The estimator of the $p$th quantile of the delta-lognormal distribution is defined by

$$\hat{q}_{p_i} = \exp(\hat{\lambda}_{p_i}), \tag{5}$$

where $\hat{\lambda}_{p_i} = \bar{Y}_{i(1)} + \Phi^{-1}\left(\dfrac{p_i - \delta'_i}{1 - \delta'_i}\right)S_{i(1)}$.

According to *Yosboonruang, Niwitpong & Niwitpong (2022)*, the variances of $\bar{Y}_{i(1)}$ and $S^2_{i(1)}$ are

$$Var\left(\bar{Y}_{i(1)}\right) = (\exp(\mu_i))^2 * \left(\frac{\sigma_i^2}{n_{i(1)}} + 1\right) * \left(\exp\left(\frac{(n_{i(1)} - 1)\sigma_i^2}{2(n_{i(1)} + 4)(n_{i(1)} - 1) + 3\sigma_i^2}\right)\right)^2$$

$$* \left(\frac{2(n_{i(1)} - 1)(\sigma_i^4/n_{i(1)})}{2(n_{i(1)} + 4)(n_{i(1)} - 1) + (6\sigma_i^4/n_{i(1)})} + 1\right)$$

$$- \left(\exp(\mu_i) * \exp\left(\frac{(n_{i(1)} - 1)\sigma_i^2}{2(n_{i(1)} + 4)(n_{i(1)} - 1) + 3\sigma_i^2}\right)\right)^2 \qquad (6)$$

and

$$Var\left(S^2_{i(1)}\right) = \left(\frac{2\sigma_i^2}{n_{i(1)}} * \exp(4\mu_i + 2\sigma_i^2)\right) * \left(2(\exp(\sigma_i^2) - 1)^2 + \sigma_i^2(2\exp(\sigma_i^2) - 1)^2\right). \qquad (7)$$

The variance of $\hat{\lambda}_{p_i}$ is

$$Var(\hat{\lambda}_{p_i}) = \hat{\lambda}_{p_i}^2(Var(\bar{Y}_{i(1)}) + z_p^2 Var(S^2_{i(1)})), \qquad (8)$$

where $z_p$ is the $100p$ percentile of the standard normal distribution and $Var(\bar{Y}_{i(1)})$ and $Var(S^2_{i(1)})$ are defined in Eqs. (6) and (7), respectively.

Therefore, the weighted average of $\hat{\lambda}_{p_i}$ based on $k$ individual sample is defined by

$$\hat{\lambda}_p = \sum_{i=1}^{k} \frac{\hat{\lambda}_{p_i}}{Var\left(\hat{\lambda}_{p_i}\right)} \bigg/ \sum_{i=1}^{k} \frac{1}{Var\left(\hat{\lambda}_{p_i}\right)}, \qquad (9)$$

where $Var\left(\hat{\lambda}_{p_i}\right)$ is defined in Eq. (8).

The common quantile of the delta-lognormal distributions is defined by

$$\hat{\theta} = \exp(\hat{\lambda}_p), \qquad (10)$$

where $\hat{\lambda}_p$ is defined in Eq. (9).

## FGCI approach

The concept of FGCI uses a fiducial generalized pivotal quantity (FGPQ) which is a subclass of GPQ.

**Definition:** Let $Y = (Y_1, Y_2, \ldots, Y_n)$ be a random variable with the probability density function $f(y; \theta)$, where $\theta = (\mu, \sigma^2, \delta')$ is a vector of unknown parameters. Let $y = (y_1, y_2, \ldots, y_n)$ be the observed value of $Y = (Y_1, Y_2, \ldots, Y_n)$. Let $R(Y; y, \theta)$ be a

function of $Y$, $y$ and $\theta$. The $R(Y; y,\theta)$ is called the FGPQ if the following two conditions are satisfied (*Weerahandi, 1993*; *Hannig et al., 2006*):

1. For a given $Y = y$, the conditional distribution of $R(Y; y,\theta)$ is free of the nuisance parameter.

2. For every $y$, the observed value of $R(Y; y, \theta)$ is the parameter of interest.

The FGPQ for $\eta = (p - \delta')/(1 - \delta')$ can be found by replacing $\delta'$ by its FGPQ.

Let $R_\mu, R_\sigma, and\ R_\eta$ be FGPQs for $\mu$, $\sigma$, and $\eta$, respectively. According to *Hasan & Krishnamoorthy (2018)*, the FGPQ for $\lambda_p$ can be expressed as follows:

$$R_{\lambda_p} = R_\mu + \Phi^{-1}(R_\eta)R_\sigma$$

$$= \bar{y}_{(1)} + \frac{Z}{U_{(1)}}\frac{s_{(1)}}{\sqrt{n_{(1)}}} + \Phi^{-1}(R_\eta)\frac{s_{(1)}}{U_{(1)}}$$

$$= \bar{y}_{(1)} + \frac{s_{(1)}}{\sqrt{n_{(1)}}}\left(\frac{Z + \Phi^{-1}(R_\eta)\sqrt{n_{(1)}}}{U_{(1)}}\right), \tag{11}$$

where $Z \sim N(0, 1)$ and $U_{(1)} \sim \sqrt{\chi^2_{n_{(1)}-1}/(n_{(1)} - 1)}$.

Let $R_{\delta'}$ be the FGPQ for $\delta'$. It can be generated by generating from uniform $(0, 1)$. The fiducial quantity $R_\eta$ must be in the interval $(0, 1)$, or equivalently,

$$0 < \frac{p - R_{\delta'}}{1 - R_{\delta'}} < 1. \tag{12}$$

To estimate the percentiles of $R_{\lambda p}$, it is enough to estimate the percentiles of the term within parentheses in Eq. (11). Therefore, the $100p$th of the delta-lognormal distribution is $(1 - \alpha)$ upper confidence limit for $\exp(\lambda_p)$.

Let $q_{pi}$ be the $p$th quantile of the delta-lognormal distribution, where $i = 1,2,\ldots,k$. It is determined by the equation $G(q_{pi}; \mu_i,\sigma^2_i,\delta'_i) = p_i$. Thus, $q_{pi} = 0$ if $p_i < \delta'_i$. For $p_i > \delta'_i$, the $q_{pi}$ is given by

$$q_{p_i} = \exp\left(\mu_i + \Phi^{-1}\left(\frac{p_i - \delta'_i}{1 - \delta'_i}\right)\sigma_i\right) = \exp(\lambda_{p_i}), \tag{13}$$

where $i = 1,2,\ldots,k$.

The FGPQ for $\lambda_{pi}$ can be expressed as follows:

$$R_{\lambda_{p_i}} = \bar{y}_{i(1)} + \frac{s_{i(1)}}{\sqrt{n_{i(1)}}}\left(\frac{Z_i + \Phi^{-1}(R_{\eta_i})\sqrt{n_{i(1)}}}{U_{i(1)}}\right), \tag{14}$$

where $Z_i \sim N(0,1)$, $U_{i(1)} \sim \sqrt{\chi^2_{n_{i(1)}-1}/(n_{i(1)} - 1)}$, and $i = 1,2,\ldots,k$.

The FGPQ for $\sigma^2_i$ is

$$R_{\sigma^2_i} = \frac{(n_{i(1)} - 1)s^2_{i(1)}}{\chi^2_{n_{i(1)}-1}}, \tag{15}$$

where $\chi^2_{ni(1)} - 1$ is chi-squared distribution with $n_{i(1)} - 1$ degrees of freedom.

The FGPQ for $\mu_i$ is

$$R_{\mu_i} = \bar{y}_{i(1)} - \frac{Z_i}{\sqrt{U_{i(1)}}} \sqrt{\frac{(n_{i(1)} - 1)s^2_{i(1)}}{n_{i(1)}}}, \tag{16}$$

where $Z_i$ denotes a standard normal distribution and $U_{i(1)}$ denotes a chi-squared distribution with $n_{i(1)} - 1$ degrees of freedom.

Let $B_{n_{i(0)}+0.5,n_{i(1)}+0.5}$ be the beta random variable with shape parameters $n_{i(0)} + 0.5$ and $n_{i(1)} + 0.5$. Suppose that $R_{\delta'_i}$ has the probability distribution of the $B_{n_{i(0)}+0.5,n_{i(1)}+0.5}$ random variable whose values are bounded above by $p$. Let $\phi$ is the standard normal distribution function. Let

$$\Phi^{-1}(R_{\eta_i}) = \Phi^{-1}\left(\frac{p_i - R_{\delta'_i}}{1 - R_{\delta'_i}}\right), \tag{17}$$

where $R_{\delta'_i} = H^{-1}(V_i H(p_i; n_{i(0)} + 0.5, n_{i(1)} + 0.5); n_{i(0)} + 0.5, n_{i(1)} + 0.5)$, $H(y; a, b)$ is the beta distribution function with shape parameters $a$ and $b$, and $V_i$ is uniform $(0,1)$.

Therefore, the FGPQ for $\lambda_{pi}$ is

$$R_{\lambda_{p_i}} = R_{\mu_i} + \frac{\sqrt{R_{\sigma^2_i}}}{\sqrt{n_{i(1)}}}\left(\frac{Z_i + \Phi^{-1}(R_{\eta_i})\sqrt{n_{i(1)}}}{\sqrt{U_{i(1)}}}\right), \tag{18}$$

where $U_{i(1)}$ denotes a chi-squared distribution with $n_{i(1)} - 1$ degrees of freedom, $R_{\sigma 2i}$ is defined in Eq. (15), $R_{\mu_i}$ is defined in Eq. (16), and $\phi^{-1}(R_{\eta_i})$ is defined in Eq. (17).

The FGPQs for $Var(\hat{\mu}_i)$ and $Var(\hat{\sigma}^2_i)$ are

$$R_{Var(\hat{\mu}_i)} = \left(\exp(R_{\mu_i})\right)^2 * \left(\frac{R_{\sigma^2_i}}{n_{i(1)}} + 1\right) * \left(\exp\left(\frac{(n_{i(1)} - 1)R_{\sigma^2_i}}{2(n_{i(1)} + 4)(n_{i(1)} - 1) + 3R_{\sigma^2_i}}\right)\right)^2$$

$$* \left(\frac{\dfrac{2(n_{i(1)} - 1)R^2_{\sigma^2_i}}{n_{i(1)}}}{2(n_{i(1)} + 4)(n_{i(1)} - 1) + \dfrac{6R^2_{\sigma^2_i}}{n_{i(1)}}} + 1\right)$$

$$- \left(\exp(R_{\mu_i}) * \exp\left(\frac{(n_{i(1)} - 1)R_{\sigma^2_i}}{2(n_{i(1)} + 4)(n_{i(1)} - 1) + 3R_{\sigma^2_i}}\right)\right)^2 \tag{19}$$

and

$$R_{Var(\hat{\sigma}^2_i)} = \left(\frac{2R_{\sigma^2_i}}{n_{i(1)}} * \exp\left(4R_{\mu_i} + 2R_{\sigma^2_i}\right)\right) * \left(2\left(\exp\left(R_{\sigma^2_i}\right) - 1\right)^2 + R_{\sigma^2_i}\left(2\exp\left(R_{\sigma^2_i}\right) - 1\right)^2\right). \tag{20}$$

Therefore, the FGPQ for $Var(\hat{\lambda}_{p_i})$ is

$$R_{Var(\hat{\lambda}_{p_i})} = R^2_{\lambda_{p_i}} R_{Var(\hat{\mu}_i)} + z^2_p R_{Var(\hat{\sigma}^2_i)}, \tag{21}$$

where $z_p$ is the $100p$ th percentile of the standard normal distribution and $R_{\lambda_{p_i}}, R_{Var(\hat{\mu}_i)}$, and $R_{Var(\hat{\sigma}^2_i)}$ are defined in Eqs. (18)–(20), respectively.

The weighted average of the FGPQ of $\lambda_{p_i}$ based on $k$ individual sample is given by

$$R_{\lambda_p} = \sum_{i=1}^{k} \frac{R_{\lambda_{p_i}}}{R_{Var(\hat{\lambda}_{p_i})}} \bigg/ \sum_{i=1}^{k} \frac{1}{R_{Var(\hat{\lambda}_{p_i})}}, \tag{22}$$

where $R_{\lambda_{p_i}}$ is defined in Eq. (18) and $R_{Var(\hat{\lambda}_{p_i})}$ is defined in Eq. (21).

The FGPQ in Eq. (22) satisfies two conditions of the definition given above. The FGCI is constructed using the quantiles of FGPQ defined in Eq. (22). Therefore, the $100(1 - \alpha)\%$ two-sided confidence interval for the weighted average of $\lambda_{p_i}$ based on the FGCI approach is

$$[L_{\lambda.FGCI}, U_{\lambda.FGCI}] = [R_{\lambda_p}(\alpha/2), R_{\lambda_p}(1 - \alpha/2)], \tag{23}$$

where $R_{\lambda_p}(\alpha/2)$ and $R_{\lambda_p}(1 - \alpha/2)$ denote the $100(\alpha/2)$-th and $100(1- \alpha/2)$-th percentiles of $R_{\lambda_p}$, respectively.

Therefore, the $100(1 - \alpha)\%$ two-sided confidence interval for the common percentile $\theta$ based on the FGCI approach is

$$CI_{FGCI} = [L_{FGCI}, U_{FGCI}] = [\exp(L_{\lambda.FGCI}), \exp(U_{\lambda.FGCI})], \tag{24}$$

where $L_{\lambda.FGCI}$ and $U_{\lambda.FGCI}$ are defined in Eq. (23).

Algorithm 1 is used to construct the FGCI:

## Adjusted MOVER approach

*Hasan & Krishnamoorthy (2018)* proposed the confidence interval for $\lambda_p$. Let $l_i$ and $u_i$ be the lower and upper limits of the confidence interval for $\lambda_{pi}$, where $i = 1,2,\dots,k$. Following *Hasan & Krishnamoorthy (2018)*, the lower and upper limits of the confidence interval are given by

$$l_i = \bar{Y}_{i(1)} - t_{n_{i(1)}-1;1-\alpha}(z_{\hat{\eta}}\sqrt{n_{i(1)}}) \frac{S_{i(1)}}{\sqrt{n_{i(1)}}} \tag{25}$$

and

$$u_i = \bar{Y}_{i(1)} + t_{n_{i(1)}-1;1-\alpha}(z_{\hat{\eta}}\sqrt{n_{i(1)}}) \frac{S_{i(1)}}{\sqrt{n_{i(1)}}}, \tag{26}$$

where $z_{\hat{\eta}}$ is the $100\,\hat{\eta}$ percentile of the standard normal distribution and $t_{n_{i(1)}-1;1-\alpha}$ $(z_{\hat{\eta}}\sqrt{n_{i(1)}})$ denotes the $q$th quantile of the noncentral $t$ distribution with $n_{i(1)} - 1$ degrees of freedom and the noncentrality parameter $z_{\hat{\eta}}\sqrt{n_{i(1)}}$.

**Algorithm 1**

For a given $\bar{y}_{i(1)}$, $s^2_{i(1)}$, and $n_{i(1)}$, where $i = 1,2,\ldots,k$

For $g = 1$ to $m$, where $m$ is number of generalized computations

Generate $\chi^2_{n_{i(1)}-1}$ from chi-squared distribution with $n_{i(1)} - 1$ degrees of freedom

Compute $R_{\sigma_i^2}$ from Eq. (15)

Compute $R_{\mu_i}$ from Eq. (16)

Compute $R_{\lambda_{p_i}}$ from Eq. (18)

Compute $R_{Var(\hat{\mu}_i)}$ from Eq. (19)

Compute $R_{Var(\hat{\sigma}_i^2)}$ from Eq. (20)

Compute $R_{Var(\hat{\lambda}_{p_i})}$ from Eq. (21)

Compute $R_{\lambda_p}$ from Eq. (22)

End $g$ loop

Compute $R_{\lambda_p}(\alpha/2)$ and $R_{\lambda_p}(1 - \alpha/2)$ from Eq. (23)

Compute $L_{FGCI}$ and $U_{FGCI}$ from Eq. (24)

Let $z_{\alpha/2}$ be the $(\alpha/2)$-th quantile of the standard normal distribution. According to *Graybill & Deal (1959)*, the $\lambda_p$ is weighted average of $\hat{\lambda}_{p_i}$ based on $k$ individual samples. Therefore, the weighted average of $\lambda_{p_i}$ is defined by

$$\hat{\lambda}_p = \sum_{i=1}^k \frac{\hat{\lambda}_{p_i}}{\widehat{Var}(\hat{\lambda}_{p_i})} \Bigg/ \sum_{i=1}^k \frac{1}{\widehat{Var}(\hat{\lambda}_{p_i})}, \tag{27}$$

where

$$\hat{\lambda}_{p_i} = \bar{Y}_{i(1)} + \Phi^{-1}\left(\frac{p_i - \delta_i'}{1 - \delta_i'}\right) S_{i(1)} \tag{28}$$

and

$$\widehat{Var}(\hat{\lambda}_{p_i}) = \frac{1}{2}\left(\frac{(\hat{\lambda}_{p_i} - l_i)^2}{z_{\alpha/2}^2} + \frac{(u_i - \hat{\lambda}_{p_i})^2}{z_{\alpha/2}^2}\right). \tag{29}$$

*Thangjai & Niwitpong (2020a)* proposed the concept of adjusted MOVER approach. The lower and upper limits of the confidence interval for the weighted average of $\lambda_{p_i}$ based on adjusted MOVER approach are defined by

$$L_{\lambda.AM} = \hat{\lambda}_p - \sqrt{\sum_{i=1}^k \frac{(\hat{\lambda}_{p_i} - l_i)^2}{(\widehat{Var}(\hat{\lambda}_{p_{l_i}}))^2} \Bigg/ \sum_{i=1}^k \frac{1}{(\widehat{Var}(\hat{\lambda}_{p_{l_i}}))^2}} \tag{30}$$

and

$$U_{\lambda.AM} = \hat{\lambda}_p + \sqrt{\sum_{i=1}^{k} \frac{(u_i - \hat{\lambda}_{p_i})^2}{(\widehat{Var}(\hat{\lambda}_{p_{u_i}}))^2} \Big/ \sum_{i=1}^{k} \frac{1}{(\widehat{Var}(\hat{\lambda}_{p_{u_i}}))^2}}, \tag{31}$$

where

$$\widehat{Var}(\hat{\lambda}_{p_{l_i}}) = \frac{(\hat{\lambda}_{p_i} - l_i)^2}{z_{\alpha/2}^2} \tag{32}$$

and

$$\widehat{Var}(\hat{\lambda}_{p_{u_i}}) = \frac{(u_i - \hat{\lambda}_{p_i})^2}{z_{\alpha/2}^2}. \tag{33}$$

Substituting $l_i$ is defined in Eq. (25) and $u_i$ is defined in Eq. (26) back into Eqs. (30) and (31), the confidence interval for the weighted average of $\lambda_{p_i}$ based on the adjusted MOVER approach is obtained.

Therefore, the $100(1 - \alpha)\%$ two-sided confidence interval for the common percentile $\theta$ based on the adjusted MOVER approach is also obtained by

$$CI_{AM} = [L_{AM}, U_{AM}] = [\exp(L_{\lambda.AM}), \exp(U_{\lambda.AM})], \tag{34}$$

where $L_{\lambda.AM}$ and $U_{\lambda.AM}$ are defined in Eqs. (30) and (31), respectively.

## Bayesian approach

Here, the Bayesian approach is used to construct two confidence intervals for the common percentile of delta-lognormal distributions. First, the Bayesian confidence interval is constructed by using fiducial quantity. Second, the Bayesian confidence interval is constructed by using an approximate fiducial distribution.

*Bayes (1763)* introduced the Bayesian approach. The Bayesian approach uses the posterior distribution for constructing the Bayesian confidence interval. The posterior distribution is a conditional distribution and uses to make statements about the parameter.

First, the Bayesian confidence interval based on fiducial quantity is considered. According to *Thangjai, Niwitpong & Niwitpong (2020b)*, the posterior distribution for $\sigma_i^2$ is inverse gamma distribution. It is defined by

$$\sigma_i^2 | y_i \sim IG\left(\frac{n_{i(1)} - 1}{2}, \frac{(n_{i(1)} - 1)s_{i(1)}^2}{2}\right). \tag{35}$$

The conditional posterior distribution for $\mu_i$ given $\sigma_i^2$ and $y_i$ is the normal distribution. It is defined by

$$\mu_i | \sigma_i^2, y_i \sim N\left(\hat{\mu}_i, \frac{\sigma_i^2}{n_{i(1)}}\right), \tag{36}$$

where $\hat{\mu}_i = \bar{y}_{i(1)}$ and $\sigma_i^2$ is defined in Eq. (35).

Let $B_{n_{i(0)}+0.5,n_{i(1)}+0.5}$ be the beta random variable with shape parameters $n_{i(0)} + 0.5$ and $n_{i(1)} + 0.5$. Also, let $Q_{\delta'i}$ be the probability distribution of the $B_{n_{i(0)}+0.5,n_{i(1)}+0.5}$ random variable. Suppose that

$$Q_{\eta_i} = \frac{p_i - Q_{\delta_i'}}{1 - Q_{\delta_i'}}, \tag{37}$$

where $Q_{\delta_i'} = H^{-1}(V_i H(p_i; n_{i(0)} + 0.5, n_{i(1)} + 0.5); n_{i(0)} + 0.5, n_{i(1)} + 0.5)$, $V_i$ denotes uniform $(0,1)$, and $H(p_i; n_{i(0)} + 0.5, n_{i(1)} + 0.5)$ denotes the beta distribution function with shape parameters $n_{i(0)} + 0.5$ and $n_{i(1)} + 0.5$.

The posterior distribution of $\lambda_{pi}$ is defined by

$$\lambda_{p_i} = \mu_i + \frac{\sqrt{\sigma_i^2}}{\sqrt{n_{i(1)}}} \left( \frac{Z_i + \Phi^{-1}(Q_{\eta_i})\sqrt{n_{i(1)}}}{\sqrt{U_{i(1)}}} \right), \tag{38}$$

where $\phi$ denotes the standard normal distribution function, $U_{i(1)}$ denotes a chi-squared distribution with $n_{i(1)} - 1$ degrees of freedom, and $\sigma_i^2$ and $\mu_i$ are defined in Eqs. (35) and (36), respectively.

The variance of $\bar{Y}_{i(1)}$ is

$$Var\left(\bar{Y}_{i(1)}\right) = (\exp(\mu_i))^2 * \left( \frac{\sigma_i^2}{n_{i(1)}} + 1 \right) * \left( \exp\left( \frac{(n_{i(1)} - 1)\sigma_i^2}{2(n_{i(1)} + 4)(n_{i(1)} - 1) + 3\sigma_i^2} \right) \right)^2$$

$$* \left( \frac{\dfrac{2(n_{i(1)} - 1)\sigma_i^4}{n_{i(1)}}}{2(n_{i(1)} + 4)(n_{i(1)} - 1) + \dfrac{6\sigma_i^4}{n_{i(1)}}} + 1 \right)$$

$$- \left( \exp(\mu_i) * \exp\left( \frac{(n_{i(1)} - 1)\sigma_i^2}{2(n_{i(1)} + 4)(n_{i(1)} - 1) + 3\sigma_i^2} \right) \right)^2, \tag{39}$$

where $\sigma_i^2$ and $\mu_i$ are defined in Eqs. (35) and (36), respectively.

The variance of $S^2_{i(1)}$ is

$$Var\left(S^2_{i(1)}\right) = \left( \left( \frac{2\sigma_i^2}{n_{i(1)}} \right) \exp\left(4\mu_i + 2\sigma_i^2\right) \right) * \left( 2\left(\exp(\sigma_i^2) - 1\right)^2 + \sigma_i^2\left(2\exp(\sigma_i^2) - 1\right)^2 \right). \tag{40}$$

Therefore, the variance of $\hat{\lambda}_{p_i}$ is

$$Var(\hat{\lambda}_{p_i}) = \lambda_{p_i}^2 Var(\bar{Y}_{i(1)}) + z_p^2 Var(S^2_{i(1)}), \tag{41}$$

where $z_p$ is the $100p$ percentile of the standard normal distribution and $\lambda_{p_i}$, $Var(\bar{Y}_{i(1)})$, and $Var(S^2_{i(1)})$ are defined in Eqs. (38)–(40), respectively.

**Algorithm 2**

For a given $\bar{y}_{i(1)}$, $s^2_{i(1)}$, and $n_{i(1)}$, where $i = 1,2,\ldots,k$

For $g = 1$ to $m$

Generate $\sigma^2_i|y_i$ from Eq. (35)

Generate $\mu_i|\sigma^2_i, y_i$ from Eq. (36)

Compute $Q_{\eta_i}$ from Eq. (37)

Compute $\lambda_{p_i}$ from Eq. (38)

Compute $Var(\bar{Y}_{i(1)})$ from Eq. (39)

Compute $Var(S^2_{i(1)})$ from Eq. (40)

Compute $Var(\hat{\lambda}_{p_i})$ from Eq. (41)

Compute $\lambda_p$ from Eq. (42)

End $g$ loop

Compute $L_{\lambda.BS1}$ and $U_{\lambda.BS1}$

Compute $L_{BS1}$ and $U_{BS1}$

Therefore, the weighted average of $\lambda_{pi}$ is defined by

$$\lambda_p = \sum_{i=1}^{k} \frac{\lambda_{p_i}}{Var\left(\hat{\lambda}_{p_i}\right)} \Bigg/ \sum_{i=1}^{k} \frac{1}{Var\left(\hat{\lambda}_{p_i}\right)}, \tag{42}$$

where $\lambda_{p_i}$ and $Var(\hat{\lambda}_{p_i})$ are defined in Eqs. (38) and (41), respectively.

According to *Gelman et al. (2013)*, the highest posterior density interval is used to construct the Bayesian confidence interval. Therefore, the confidence interval for the weighted average of $\lambda_{p_i}$ based on Bayesian approach using fiducial quantity are defined by

$$CI_{\lambda.BS1} = [L_{\lambda.BS1}, U_{\lambda.BS1}], \tag{43}$$

where $L_{\lambda.BS1}$ and $U_{\lambda.BS1}$ are the lower and upper limits of the shortest $100(1-\alpha)\%$ highest posterior density interval of $\lambda_p$, respectively.

Therefore, the $100(1-\alpha)\%$ two-sided confidence interval for the common percentile $\theta$ based on the Bayesian approach using fiducial quantity is also obtained by

$$CI_{BS1} = [L_{BS1}, U_{BS1}] = [\exp(L_{\lambda.BS1}), \exp(U_{\lambda.BS1})], \tag{44}$$

where $L_{\lambda.BS1}$ and $U_{\lambda.BS1}$ are defined in Eq. (43).

The Algorithm 2 is used to construct the Bayesian confidence interval using fiducial quantity:

Second, the Bayesian confidence interval using an approximate fiducial distribution is considered. The posterior distribution of $\lambda_{p_i}$ is defined by

$$\lambda_{p_i} = \mu_i + \frac{\sqrt{\sigma^2_i}}{\sqrt{n_{i(1)}}} t_{n_{i(1)}-1;1-\alpha}\left(z_{\hat{\eta}}\sqrt{n_{i(1)}}\right), \tag{45}$$

**Algorithm 3**

For a given $\bar{y}_{i(1)}$, $s^2_{i(1)}$, and $n_{i(1)}$, where $i = 1,2,\ldots,k$

For $g = 1$ to $m$

Generate $\sigma_i^2|y_i$ from Eq. (35)

Generate $\mu_i|\sigma_i^2,y_i$ from Eq. (36)

Compute $\lambda_{p_i}$ from Eq. (45)

Compute $Var(\bar{Y}_{i(1)})$ from Eq. (39)

Compute $Var(S^2_{i(1)})$ from Eq. (40)

Compute $Var(\hat{\lambda}_{p_i})$ from Eq. (46)

Compute $\lambda_p$ from Eq. (47)

End $g$ loop

Compute $L_{\lambda.BS2}$ and $U_{\lambda.BS2}$

Compute $L_{BS2}$ and $U_{BS2}$

where $\sigma_i^2$ and $\mu_i$ are defined in Eqs. (35) and (36), respectively, $t_{n_{i(1)}-1;1-\alpha}(z_{\hat{\eta}}\sqrt{n_{i(1)}})$ is the $(1-\alpha)$th quantile of the noncentral $t$ distribution with $n_{i(1)} - 1$ degrees of freedom and the noncentrality parameter $z_{\hat{\eta}}\sqrt{n_{i(1)}}$.

Therefore, the variance of $\hat{\lambda}_{p_i}$ is

$$Var(\hat{\lambda}_{p_i}) = \lambda_{p_i}^2 Var(\bar{Y}_{i(1)}) + z_p^2 Var(S^2_{i(1)}), \tag{46}$$

where $z_p$ is the 100p percentile of the standard normal distribution and $Var(\bar{Y}_{i(1)})$, $Var(S^2_{i(1)})$, and $\lambda_{p_i}$ are defined in Eqs. (39), (40) and (45), respectively.

Therefore, the weighted average of $\lambda_{p_i}$ is defined by

$$\lambda_p = \sum_{i=1}^{k} \frac{\lambda_{p_i}}{Var\left(\hat{\lambda}_{p_i}\right)} \bigg/ \sum_{i=1}^{k} \frac{1}{Var\left(\hat{\lambda}_{p_i}\right)}, \tag{47}$$

where $\lambda_{p_i}$ and $Var(\hat{\lambda}_{p_i})$ are defined in Eqs. (45) and (46), respectively.

Therefore, the confidence interval for the weighted average of $\lambda_{pi}$ based on Bayesian approach using an approximate fiducial distribution are defined by

$$CI_{\lambda.BS2} = [L_{\lambda.BS2}, U_{\lambda.BS2}], \tag{48}$$

where $L_{\lambda.BS2}$ and $U_{\lambda.BS2}$ are the lower and upper limits of the shortest $100(1-\alpha)\%$ highest posterior density interval of $\lambda_p$, respectively.

Therefore, the $100(1-\alpha)\%$ two-sided confidence interval for the common percentile $\theta$ based on the Bayesian approach using an approximate fiducial distribution is also obtained by

$$CI_{BS2} = [L_{BS2}, U_{BS2}] = [\exp(L_{\lambda.BS2}), \exp(U_{\lambda.BS2})], \tag{49}$$

where $L_{\lambda.BS2}$ and $U_{\lambda.BS2}$ are defined in Eq. (48).

**Algorithm 4.**

For a given $(n_1, n_2, \ldots, n_k)$, $(\mu_1, \mu_2, \ldots, \mu_k)$, $(\sigma_1^2, \sigma_2^2, \ldots, \sigma_k^2)$, $(\delta_1', \delta_2', \ldots, \delta_k')$ and $\theta$

For $h = 1$ to $M$

Generate $x_{ij}$ from $DLN(\mu_i, \sigma_i^2, \delta_i')$, where $i = 1, 2, \ldots, k$ and $j = 1, 2, \ldots, n_i$

Compute $y_{ij} = ln(x_{ij})$, where $i = 1, 2, \ldots, k$ and $j = 1, 2, \ldots, n_i$

Compute $\bar{y}_{i(1)}$, $s^2_{i(1)}$, and $n_{i(1)}$, where $i = 1, 2, \ldots, k$

Use Algorithm 1 with $m = 1,000$ to construct $CI_{FGCI(h)} = [L_{FGCI(h)}, U_{FGCI(h)}]$

Use Eq. (34) to construct $CI_{AM(h)} = [L_{AM(h)}, U_{AM(h)}]$

Use Algorithm 2 with $m = 1,000$ to construct $CI_{BS1(h)} = [L_{BS1(h)}, U_{BS1(h)}]$

Use Algorithm 3 with $m = 1,000$ to construct $CI_{BS2(h)} = [L_{BS2(h)}, U_{BS2(h)}]$

If $[L_{(h)} \leqslant \theta \leqslant U_{(h)}]$, set $p_{(h)} = 1$; else set $p_{(h)} = 0$

Compute $U_{(h)} - L_{(h)}$

End $h$ loop

Compute mean of $p_{(h)}$ defined by the coverage probability

Compute mean of $U_{(h)} - L_{(h)}$ defined by the average length

The Algorithm 3 is used to construct the Bayesian confidence interval using an approximate fiducial distribution:

## RESULTS

FGCI, adjusted MOVER, and two Bayesian approaches were used to construct four confidence intervals for the common percentile of delta-lognormal distributions denoted as $CI_{FGCI}$, $CI_{AM}$, and $CI_{BS1}$ and $CI_{BS2}$, respectively. The performances of the confidence intervals were compared by computing their coverage probabilities and average lengths.

The coverage probability of the $100(1 - \alpha)\%$ confidence level is $c \pm z_{\alpha/2}\sqrt{\dfrac{c(1 - c)}{M}}$, where $c$ is the nominal confidence level, $z_{\alpha/2}$ is the $(\alpha/2)$-th quantile of a standard normal distribution, and $M$ is the number of simulation runs. At the 95% confidence level, the most appropriate confidence interval will have a coverage probability in the range [0.9440,0.9560] with the shortest average length. For the FGCI and Bayesian approaches, the critical value was evaluated for 1,000 samples. The simulation results were obtained from 5,000 runs.

The Algorithm 4 is used to evaluate the coverage probability and average length:

In this simulation study, the nominal confidence level was chosen as 0.95. The variables in the simulation study were set as number of samples $k = 3$ or $k = 6$; population means $\mu_1 = \mu_2 = \ldots = \mu_k = 1$; population variances $\sigma_1^2, \sigma_2^2, \ldots, \sigma_k^2$; the probabilities of obtaining zero observations $\delta_1', \delta_2', \ldots, \delta_k'$; and sample sizes $n_1, n_2, \ldots, n_k$.

The performances of the proposed confidence intervals for $k = 3$ and $k = 6$ are given in Tables 1 and 2 and displayed in Figs. 1–6. For $k = 3$, the coverage probabilities of $CI_{FGCI}$ were greater than the nominal confidence level of 0.95 for all cases whereas those of $CI_{AM}$ were less than 0.95 when the sample sizes were small but close otherwise. For the Bayesian

**Table 1 The coverage probabilities (CPs) and average lengths (ALs) of 95% two-sided confidence intervals for the common percentile of several delta-lognormal populations: Three sample cases.**

| $(n_1, n_2, n_3)$ | $(\mu_1, \mu_2, \mu_3)$ | $(\delta'_1, \delta'_2, \delta'_3)$ | $(\sigma^2_1, \sigma^2_2, \sigma^2_3)$ | | CP (AL) | | |
|---|---|---|---|---|---|---|---|
| | | | | $CI_{FGCI}$ | $CI_{AM}$ | $CI_{BS1}$ | $CI_{BS2}$ |
| (10, 10, 10) | (1, 1, 1) | (0.1, 0.1, 0.1) | (0.5, 0.5, 0.5) | 0.9616 | 0.9188 | 0.9302 | 0.9122 |
| | | | | (13.1225) | (13.5253) | (11.1999) | (16.5684) |
| | | | (0.5, 0.5, 1) | 0.9788 | 0.9464 | 0.9614 | 0.8578 |
| | | | | (16.7407) | (16.1035) | (14.1176) | (21.7400) |
| | | | (1, 1, 1) | 0.9656 | 0.9276 | 0.9362 | 0.9108 |
| | | | | (35.3099) | (29.9111) | (28.3798) | (49.9308) |
| | | | (1, 1, 2) | 0.9794 | 0.9480 | 0.9602 | 0.8580 |
| | | | | (46.7255) | (36.5878) | (36.8141) | (67.6506) |
| | | | (2, 2, 2) | 0.9684 | 0.9238 | 0.9428 | 0.9240 |
| | | | | (129.2134) | (84.0922) | (93.1638) | (201.9722) |
| | | (0.1, 0.1, 0.3) | (0.5, 0.5, 0.5) | 0.9626 | 0.9180 | 0.9352 | 0.8906 |
| | | | | (14.0448) | (13.4411) | (11.9074) | (18.7175) |
| | | | (0.5, 0.5, 1) | 0.9754 | 0.9248 | 0.9550 | 0.8696 |
| | | | | (16.9407) | (15.0689) | (14.2855) | (22.2965) |
| | | | (1, 1, 1) | 0.9650 | 0.9104 | 0.9366 | 0.9036 |
| | | | | (36.3109) | (28.2836) | (28.9786) | (55.5862) |
| | | | (1, 1, 2) | 0.9760 | 0.9252 | 0.9550 | 0.8694 |
| | | | | (47.6926) | (33.6305) | (37.6027) | (71.4143) |
| | | | (2, 2, 2) | 0.9656 | 0.9140 | 0.9406 | 0.8912 |
| | | | | (148.9675) | (82.9091) | (103.9126) | (270.3032) |
| | | (0.3, 0.3, 0.3) | (0.5, 0.5, 0.5) | 0.9656 | 0.8762 | 0.9404 | 0.9046 |
| | | | | (15.5403) | (12.1970) | (12.8115) | (19.1430) |
| | | | (0.5, 0.5, 1) | 0.9772 | 0.9066 | 0.9586 | 0.8540 |
| | | | | (20.0826) | (14.4529) | (16.1167) | (25.2621) |
| | | | (1, 1, 1) | 0.9658 | 0.8752 | 0.9418 | 0.9110 |
| | | | | (42.2880) | (25.1633) | (31.9548) | (56.9812) |
| | | | (1, 1, 2) | 0.9800 | 0.9082 | 0.9662 | 0.8656 |
| | | | | (59.9388) | (31.0398) | (43.3691) | (83.2374) |
| | | | (2, 2, 2) | 0.9672 | 0.8722 | 0.9356 | 0.9114 |
| | | | | (190.4600) | (70.9535) | (118.5987) | (269.8711) |
| | | (0.3, 0.3, 0.5) | (0.5, 0.5, 0.5) | 0.9672 | 0.8810 | 0.9456 | 0.8878 |
| | | | | (15.8362) | (11.7858) | (12.9445) | (19.0324) |
| | | | (0.5, 0.5, 1) | 0.9758 | 0.8934 | 0.9584 | 0.8794 |
| | | | | (20.2488) | (13.5859) | (16.2743) | (24.4542) |
| | | | (1, 1, 1) | 0.9730 | 0.8826 | 0.9534 | 0.8912 |
| | | | | (43.8636) | (24.3271) | (32.7406) | (57.1125) |
| | | | (1, 1, 2) | 0.9802 | 0.8916 | 0.9596 | 0.8790 |
| | | | | (60.3466) | (29.5326) | (43.3793) | (79.5503) |
| | | | (2, 2, 2) | 0.9732 | 0.8810 | 0.9510 | 0.8938 |
| | | | | (197.5581) | (67.5715) | (122.5223) | (268.6405) |

| $(n_1, n_2, n_3)$ | $(\mu_1, \mu_2, \mu_3)$ | $(\delta'_1, \delta'_2, \delta'_3)$ | $(\sigma^2_1, \sigma^2_2, \sigma^2_3)$ | $CI_{FGCI}$ | CP (AL) $CI_{AM}$ | $CI_{BS1}$ | $CI_{BS2}$ |
|---|---|---|---|---|---|---|---|
| | | (0.5, 0.5, 0.5) | (0.5, 0.5, 0.5) | 0.9774 | 0.8736 | 0.9588 | 0.8836 |
| | | | | (16.7548) | (10.9530) | (13.3976) | (17.8895) |
| | | | (0.5, 0.5, 1) | 0.9864 | 0.9016 | 0.9732 | 0.8486 |
| | | | | (21.9578) | (13.0428) | (17.0648) | (23.6756) |
| | | | (1, 1, 1) | 0.9780 | 0.8732 | 0.9562 | 0.8874 |
| | | | | (48.3134) | (22.2962) | (34.3590) | (52.5244) |
| | | | (1, 1, 2) | 0.9848 | 0.8948 | 0.9684 | 0.8584 |
| | | | | (70.4925) | (27.6249) | (47.5502) | (75.9134) |
| | | | (2, 2, 2) | 0.9774 | 0.8676 | 0.9552 | 0.8978 |
| | | | | (227.3056) | (58.6553) | (129.4709) | (233.7485) |
| | | (0.1, 0.3, 0.5) | (0.5, 0.5, 0.5) | 0.9712 | 0.9148 | 0.9508 | 0.8554 |
| | | | | (15.2138) | (12.7495) | (12.6556) | (20.9990) |
| | | | (0.5, 0.5, 1) | 0.9744 | 0.9112 | 0.9568 | 0.8492 |
| | | | | (18.5054) | (14.1463) | (15.3719) | (25.5513) |
| | | | (1, 1, 1) | 0.9738 | 0.9138 | 0.9522 | 0.8634 |
| | | | | (40.5824) | (26.4336) | (31.3253) | (65.7662) |
| | | | (1, 1, 2) | 0.9758 | 0.9142 | 0.9598 | 0.8592 |
| | | | | (53.6772) | (31.1696) | (40.8956) | (85.5507) |
| | | | (2, 2, 2) | 0.9726 | 0.9078 | 0.9466 | 0.8586 |
| | | | | (166.2144) | (74.6314) | (111.7168) | (316.6658) |
| (30, 30, 30) | (1, 1, 1) | (0.1, 0.1, 0.1) | (0.5, 0.5, 0.5) | 0.9616 | 0.9998 | 0.9480 | 0.3428 |
| | | | | (6.5534) | (15.5421) | (6.2191) | (11.3560) |
| | | | (0.5, 0.5, 1) | 0.9832 | 0.9994 | 0.9764 | 0.2838 |
| | | | | (7.7955) | (18.0345) | (7.3614) | (13.9841) |
| | | | (1, 1, 1) | 0.9662 | 0.9999 | 0.9500 | 0.3958 |
| | | | | (14.7780) | (33.5906) | (13.9057) | (31.7637) |
| | | | (1, 1, 2) | 0.9842 | 0.9996 | 0.9768 | 0.2916 |
| | | | | (18.1783) | (41.1211) | (16.9299) | (40.8851) |
| | | | (2, 2, 2) | 0.9682 | 0.9996 | 0.9474 | 0.3946 |
| | | | | (41.9636) | (98.4243) | (38.5704) | (121.0134) |
| | | (0.1, 0.1, 0.3) | (0.5, 0.5, 0.5) | 0.9664 | 0.9992 | 0.9538 | 0.2796 |
| | | | | (6.6627) | (15.3912) | (6.3104) | (12.6293) |
| | | | (0.5, 0.5, 1) | 0.9798 | 0.9996 | 0.9696 | 0.3076 |
| | | | | (7.7158) | (16.9017) | (7.2933) | (14.1194) |
| | | | (1, 1, 1) | 0.9622 | 0.9994 | 0.9478 | 0.3154 |
| | | | | (14.7796) | (32.9862) | (13.8794) | (36.2047) |
| | | | (1, 1, 2) | 0.9842 | 0.9994 | 0.9750 | 0.3334 |
| | | | | (18.0534) | (37.8567) | (16.8752) | (41.6387) |
| | | | (2, 2, 2) | 0.9740 | 0.9996 | 0.9556 | 0.3324 |
| | | | | (41.3229) | (96.0415) | (38.0324) | (146.8337) |
| | | (0.3, 0.3, 0.3) | (0.5, 0.5, 0.5) | 0.9626 | 0.9984 | 0.9472 | 0.4062 |

(Continued)

| | | | | | CP (AL) | | |
|---|---|---|---|---|---|---|---|
| $(n_1, n_2, n_3)$ | $(\mu_1, \mu_2, \mu_3)$ | $(\delta'_1, \delta'_2, \delta'_3)$ | $(\sigma^2_1, \sigma^2_2, \sigma^2_3)$ | $CI_{FGCI}$ | $CI_{AM}$ | $CI_{BS1}$ | $CI_{BS2}$ |
| | | | | (6.6455) | (14.1889) | (6.2642) | (11.9336) |
| | | | (0.5, 0.5, 1) | 0.9822 | 0.9996 | 0.9728 | 0.3346 |
| | | | | (7.8844) | (16.4172) | (7.3760) | (14.7705) |
| | | | (1, 1, 1) | 0.9620 | 0.9978 | 0.9472 | 0.4338 |
| | | | | (14.5852) | (29.7677) | (13.5198) | (32.8589) |
| | | | (1, 1, 2) | 0.9834 | 0.9992 | 0.9714 | 0.3410 |
| | | | | (17.9405) | (36.5177) | (16.5369) | (43.0495) |
| | | | (2, 2, 2) | 0.9656 | 0.9984 | 0.9480 | 0.4668 |
| | | | | (40.0606) | (83.4089) | (36.2743) | (125.1631) |
| | | (0.3, 0.3, 0.5) | (0.5, 0.5, 0.5) | 0.9706 | 0.9992 | 0.9570 | 0.3560 |
| | | | | (6.6983) | (13.8889) | (6.2906) | (12.5125) |
| | | | (0.5, 0.5, 1) | 0.9756 | 0.9972 | 0.9660 | 0.3978 |
| | | | | (7.7910) | (15.2157) | (7.3026) | (14.3137) |
| | | | (1, 1, 1) | 0.9692 | 0.9992 | 0.9556 | 0.3772 |
| | | | | (14.5416) | (28.9139) | (13.4969) | (35.2043) |
| | | | (1, 1, 2) | 0.9814 | 0.9986 | 0.9724 | 0.4040 |
| | | | | (17.9096) | (33.2614) | (16.5173) | (41.8378) |
| | | | (2, 2, 2) | 0.9766 | 0.9992 | 0.9596 | 0.3980 |
| | | | | (39.5256) | (79.6347) | (35.6352) | (136.1955) |
| | | (0.5, 0.5, 0.5) | (0.5, 0.5, 0.5) | 0.9776 | 0.9974 | 0.9678 | 0.4478 |
| | | | | (6.7086) | (12.7519) | (6.2497) | (11.3794) |
| | | | (0.5, 0.5, 1) | 0.9876 | 0.9990 | 0.9818 | 0.3698 |
| | | | | (7.8981) | (14.6321) | (7.3269) | (13.9327) |
| | | | (1, 1, 1) | 0.9808 | 0.9978 | 0.9704 | 0.4718 |
| | | | | (14.4210) | (25.9801) | (13.2412) | (30.2652) |
| | | | (1, 1, 2) | 0.9906 | 0.9986 | 0.9856 | 0.3730 |
| | | | | (17.8024) | (31.4804) | (16.1841) | (39.4965) |
| | | | (2, 2, 2) | 0.9822 | 0.9978 | 0.9694 | 0.4786 |
| | | | | (38.0542) | (67.9039) | (33.8151) | (106.6205) |
| | | (0.1, 0.3, 0.5) | (0.5, 0.5, 0.5) | 0.9742 | 0.9996 | 0.9646 | 0.1876 |
| | | | | (6.8272) | (14.8090) | (6.4405) | (14.7957) |
| | | | (0.5, 0.5, 1) | 0.9818 | 0.9992 | 0.9720 | 0.2618 |
| | | | | (7.8849) | (16.0906) | (7.4282) | (16.3482) |
| | | | (1, 1, 1) | 0.9764 | 0.9996 | 0.9590 | 0.2104 |
| | | | | (14.9332) | (31.3658) | (13.9415) | (44.6454) |
| | | | (1, 1, 2) | 0.9812 | 0.9996 | 0.9710 | 0.2754 |
| | | | | (18.1263) | (35.2511) | (16.9290) | (50.0834) |
| | | | (2, 2, 2) | 0.9764 | 0.9998 | 0.9650 | 0.2354 |
| | | | | (40.6140) | (88.0470) | (37.2159) | (189.3741) |
| (50, 50, 50) | (1, 1, 1) | (0.1, 0.1, 0.1) | (0.5, 0.5, 0.5) | 0.9638 | 0.9999 | 0.9546 | 0.0812 |
| | | | | (5.0444) | (15.8510) | (4.8682) | (9.2761) |

| $(n_1, n_2, n_3)$ | $(\mu_1, \mu_2, \mu_3)$ | $(\delta'_1, \delta'_2, \delta'_3)$ | $(\sigma^2_1, \sigma^2_2, \sigma^2_3)$ | | CP (AL) | | |
|---|---|---|---|---|---|---|---|
| | | | | $CI_{FGCI}$ | $CI_{AM}$ | $CI_{BS1}$ | $CI_{BS2}$ |
| | | | (0.5, 0.5, 1) | 0.9878 | 0.9999 | 0.9836 | 0.0626 |
| | | | | (5.9532) | (18.3695) | (5.7275) | (11.2863) |
| | | | (1, 1, 1) | 0.9646 | 0.9999 | 0.9538 | 0.0978 |
| | | | | (11.3314) | (34.3475) | (10.8900) | (26.3688) |
| | | | (1, 1, 2) | 0.9842 | 0.9999 | 0.9770 | 0.0764 |
| | | | | (13.5810) | (41.7624) | (13.0107) | (32.6945) |
| | | | (2, 2, 2) | 0.9660 | 0.9999 | 0.9570 | 0.1064 |
| | | | | (31.0233) | (100.0403) | (29.5751) | (99.6549) |
| | (0.1, 0.1, 0.3) | | (0.5, 0.5, 0.5) | 0.9578 | 0.9999 | 0.9514 | 0.0614 |
| | | | | (5.1143) | (15.7165) | (4.9284) | (10.3754) |
| | | | (0.5, 0.5, 1) | 0.9852 | 0.9999 | 0.9792 | 0.0848 |
| | | | | (5.8783) | (17.2641) | (5.6547) | (11.3551) |
| | | | (1, 1, 1) | 0.9626 | 0.9999 | 0.9506 | 0.0690 |
| | | | | (11.3605) | (34.0661) | (10.9094) | (30.4236) |
| | | | (1, 1, 2) | 0.9816 | 0.9999 | 0.9740 | 0.0922 |
| | | | | (13.4399) | (38.3581) | (12.8614) | (32.8323) |
| | | | (2, 2, 2) | 0.9674 | 0.9999 | 0.9524 | 0.0748 |
| | | | | (30.8706) | (99.3191) | (29.4213) | (122.6326) |
| | (0.3, 0.3, 0.3) | | (0.5, 0.5, 0.5) | 0.9642 | 0.9999 | 0.9542 | 0.1258 |
| | | | | (5.0770) | (14.5207) | (4.8695) | (9.6186) |
| | | | (0.5, 0.5, 1) | 0.9852 | 0.9998 | 0.9804 | 0.1042 |
| | | | | (5.9510) | (16.7797) | (5.6947) | (11.7019) |
| | | | (1, 1, 1) | 0.9634 | 0.9999 | 0.9518 | 0.1432 |
| | | | | (11.0306) | (30.7406) | (10.5358) | (26.8617) |
| | | | (1, 1, 2) | 0.9860 | 0.9999 | 0.9774 | 0.1090 |
| | | | | (13.2021) | (37.2641) | (12.5360) | (33.7078) |
| | | | (2, 2, 2) | 0.9654 | 0.9999 | 0.9530 | 0.1632 |
| | | | | (28.8237) | (85.2569) | (27.2393) | (98.3914) |
| | (0.3, 0.3, 0.5) | | (0.5, 0.5, 0.5) | 0.9722 | 0.9999 | 0.9624 | 0.0854 |
| | | | | (5.1636) | (14.3764) | (4.9468) | (10.4763) |
| | | | (0.5, 0.5, 1) | 0.9824 | 0.9999 | 0.9758 | 0.1360 |
| | | | | (5.9178) | (15.6579) | (5.6638) | (11.6016) |
| | | | (1, 1, 1) | 0.9744 | 0.9999 | 0.9610 | 0.1060 |
| | | | | (10.9913) | (29.9042) | (10.4848) | (29.3116) |
| | | | (1, 1, 2) | 0.9828 | 0.9999 | 0.9758 | 0.1372 |
| | | | | (13.1756) | (33.8248) | (12.5288) | (33.0561) |
| | | | (2, 2, 2) | 0.9754 | 0.9999 | 0.9658 | 0.1096 |
| | | | | (28.6218) | (83.3013) | (26.9484) | (112.8997) |
| | (0.5, 0.5, 0.5) | | (0.5, 0.5, 0.5) | 0.9806 | 0.9999 | 0.9738 | 0.1624 |
| | | | | (5.0797) | (13.0450) | (4.8474) | (9.3196) |
| | | | (0.5, 0.5, 1) | 0.9888 | 0.9999 | 0.9826 | 0.1290 |

(Continued)

| $(n_1, n_2, n_3)$ | $(\mu_1, \mu_2, \mu_3)$ | $(\delta^{'}_1, \delta^{'}_2, \delta^{'}_3)$ | $(\sigma^2_1, \sigma^2_2, \sigma^2_3)$ | | CP (AL) | | |
|---|---|---|---|---|---|---|---|
| | | | | $CI_{FGCI}$ | $CI_{AM}$ | $CI_{BS1}$ | $CI_{BS2}$ |
| | | | | (5.9101) | (14.9031) | (5.6202) | (11.2249) |
| | | | (1, 1, 1) | 0.9800 | 0.9998 | 0.9690 | 0.1740 |
| | | | | (10.6122) | (26.4904) | (10.0587) | (24.5042) |
| | | | (1, 1, 2) | 0.9914 | 0.9999 | 0.9868 | 0.1364 |
| | | | | (12.7123) | (31.7257) | (11.9502) | (30.8441) |
| | | | (2, 2, 2) | 0.9832 | 0.9998 | 0.9746 | 0.1872 |
| | | | | (27.0649) | (70.5521) | (25.3041) | (86.2946) |
| | | (0.1, 0.3, 0.5) | (0.5, 0.5, 0.5) | 0.9826 | 0.9999 | 0.9752 | 0.0226 |
| | | | | (5.2820) | (15.2200) | (5.0773) | (12.6177) |
| | | | (0.5, 0.5, 1) | 0.9836 | 0.9999 | 0.9782 | 0.0502 |
| | | | | (5.9733) | (16.4098) | (5.7396) | (13.2880) |
| | | | (1, 1, 1) | 0.9756 | 0.9999 | 0.9686 | 0.0300 |
| | | | | (11.4649) | (32.4412) | (10.9935) | (38.9225) |
| | | | (1, 1, 2) | 0.9826 | 0.9999 | 0.9764 | 0.0622 |
| | | | | (13.5286) | (35.7961) | (12.9186) | (40.2835) |
| | | | (2, 2, 2) | 0.9772 | 0.9999 | 0.9642 | 0.0326 |
| | | | | (29.9892) | (91.7238) | (28.5242) | (168.0056) |
| (100, 100, 100) | (1, 1, 1) | (0.1, 0.1, 0.1) | (0.5, 0.5, 0.5) | 0.9714 | 0.9999 | 0.9700 | 0.0012 |
| | | | | (3.5528) | (16.0693) | (3.4567) | (6.8105) |
| | | | (0.5, 0.5, 1) | 0.9862 | 0.9999 | 0.9810 | 0.0022 |
| | | | | (4.1673) | (18.4836) | (4.0644) | (8.1855) |
| | | | (1, 1, 1) | 0.9596 | 0.9999 | 0.9600 | 0.0030 |
| | | | | (8.0642) | (34.8008) | (7.8775) | (20.0550) |
| | | | (1, 1, 2) | 0.9880 | 0.9999 | 0.9826 | 0.0008 |
| | | | | (9.4937) | (42.1608) | (9.2431) | (23.9587) |
| | | | (2, 2, 2) | 0.9698 | 0.9999 | 0.9628 | 0.0042 |
| | | | | (22.0811) | (101.6958) | (21.4862) | (77.4539) |
| | | (0.1, 0.1, 0.3) | (0.5, 0.5, 0.5) | 0.9724 | 0.9999 | 0.9690 | 0.0002 |
| | | | | (3.6095) | (16.0508) | (3.5168) | (7.7891) |
| | | | (0.5, 0.5, 1) | 0.9896 | 0.9999 | 0.9844 | 0.0014 |
| | | | | (4.1229) | (17.4908) | (4.0160) | (8.2179) |
| | | | (1, 1, 1) | 0.9726 | 0.9999 | 0.9684 | 0.0012 |
| | | | | (8.1324) | (34.6864) | (7.9376) | (23.5548) |
| | | | (1, 1, 2) | 0.9856 | 0.9999 | 0.9826 | 0.0014 |
| | | | | (9.4391) | (39.1547) | (9.1788) | (23.8943) |
| | | | (2, 2, 2) | 0.9730 | 0.9999 | 0.9684 | 0.0020 |
| | | | | (21.9251) | (101.2034) | (21.4158) | (96.7689) |
| | | (0.3, 0.3, 0.3) | (0.5, 0.5, 0.5) | 0.9720 | 0.9999 | 0.9684 | 0.0040 |
| | | | | (3.5729) | (14.7932) | (3.4684) | (7.0743) |
| | | | (0.5, 0.5, 1) | 0.9858 | 0.9999 | 0.9836 | 0.0056 |
| | | | | (4.1667) | (16.9502) | (4.0517) | (8.4757) |
| Table 1 (continued) | | | | | | | |
|---|---|---|---|---|---|---|---|
| $(n_1, n_2, n_3)$ | $(\mu_1, \mu_2, \mu_3)$ | $(\delta'_1, \delta'_2, \delta'_3)$ | $(\sigma^2_1, \sigma^2_2, \sigma^2_3)$ | | CP (AL) | | |
| | | | | $CI_{FGCI}$ | $CI_{AM}$ | $CI_{BS1}$ | $CI_{BS2}$ |
| | | | (1, 1, 1) | 0.9652 | 0.9999 | 0.9590 | 0.0068 |
| | | | | (7.7922) | (31.1248) | (7.5794) | (20.0571) |
| | | | (1, 1, 2) | 0.9874 | 0.9999 | 0.9822 | 0.0044 |
| | | | | (9.1701) | (37.4998) | (8.9057) | (24.2793) |
| | | | (2, 2, 2) | 0.9668 | 0.9999 | 0.9584 | 0.0070 |
| | | | | (20.1903) | (86.8276) | (19.5881) | (74.1374) |
| | (0.3, 0.3, 0.5) | (0.5, 0.5, 0.5) | | 0.9806 | 0.9999 | 0.9772 | 0.0012 |
| | | | | (3.6457) | (14.6436) | (3.5429) | (7.9609) |
| | | | (0.5, 0.5, 1) | 0.9856 | 0.9999 | 0.9830 | 0.0060 |
| | | | | (4.1307) | (15.8826) | (4.0104) | (8.4596) |
| | | | (1, 1, 1) | 0.9786 | 0.9999 | 0.9762 | 0.0024 |
| | | | | (7.8435) | (30.6015) | (7.6256) | (23.0574) |
| | | | (1, 1, 2) | 0.9840 | 0.9999 | 0.9796 | 0.0056 |
| | | | | (9.0767) | (34.0380) | (8.8051) | (23.6508) |
| | | | (2, 2, 2) | 0.9750 | 0.9999 | 0.9694 | 0.0040 |
| | | | | (20.0959) | (85.2629) | (19.5149) | (90.5855) |
| | (0.5, 0.5, 0.5) | (0.5, 0.5, 0.5) | | 0.9862 | 0.9999 | 0.9836 | 0.0112 |
| | | | | (3.5578) | (13.1718) | (3.4480) | (6.9213) |
| | | | (0.5, 0.5, 1) | 0.9916 | 0.9999 | 0.9898 | 0.0098 |
| | | | | (4.1245) | (14.9820) | (3.9938) | (8.2568) |
| | | | (1, 1, 1) | 0.9826 | 0.9999 | 0.9800 | 0.0120 |
| | | | | (7.4301) | (26.7038) | (7.2045) | (18.5641) |
| | | | (1, 1, 2) | 0.9916 | 0.9999 | 0.9874 | 0.0102 |
| | | | | (8.6974) | (31.8035) | (8.3988) | (22.4537) |
| | | | (2, 2, 2) | 0.9856 | 0.9999 | 0.9806 | 0.0130 |
| | | | | (18.2724) | (70.2128) | (17.6212) | (64.1559) |
| | (0.1, 0.3, 0.5) | (0.5, 0.5, 0.5) | | 0.9850 | 0.9999 | 0.9838 | 0.0002 |
| | | | | (3.7613) | (15.5088) | (3.6583) | (9.8258) |
| | | | (0.5, 0.5, 1) | 0.9896 | 0.9999 | 0.9850 | 0.0008 |
| | | | | (4.1564) | (16.6099) | (4.0451) | (9.7186) |
| | | | (1, 1, 1) | 0.9804 | 0.9999 | 0.9788 | 0.0004 |
| | | | | (8.2611) | (32.8864) | (8.0504) | (31.4164) |
| | | | (1, 1, 2) | 0.9838 | 0.9999 | 0.9812 | 0.0012 |
| | | | | (9.4614) | (36.4256) | (9.2045) | (29.7488) |
| | | | (2, 2, 2) | 0.9802 | 0.9999 | 0.9750 | 0.0004 |
| | | | | (21.6124) | (94.5200) | (21.1194) | (143.6208) |

methods, the coverage probabilities of $CI_{BS1}$ were greater than 0.95 for some cases whereas those of $CI_{BS2}$ were less than 0.95 for all cases. Meanwhile, the average lengths of were shorter than those of $CI_{FGCI}$.

**Table 2 The coverage probabilities and average lengths of 95% two-sided confidence intervals for the common percentile of several delta-lognormal populations: 6 sample cases.**

| $(n_1, n_2, n_3, n_4, n_5, n_6)$ | $(\mu_1, \mu_2, \mu_3, \mu_4, \mu_5, \mu_6)$ | $(\delta'_1, \delta'_2, \delta'_3, \delta'_4, \delta'_5, \delta'_6)$ | $(\sigma^2_1, \sigma^2_2, \sigma^2_3, \sigma^2_4, \sigma^2_5, \sigma^2_6)$ | CP (AL) | | | |
|---|---|---|---|---|---|---|---|
| | | | | $CI_{FGCI}$ | $CI_{AM}$ | $CI_{BS1}$ | $CI_{BS2}$ |
| (10, 10, 10, 10, 10, 10) | (1, 1, 1, 1, 1, 1) | (0.1, 0.1, 0.1, 0.1, 0.1, 0.1) | (0.5, 0.5, 0.5, 0.5, 0.5, 0.5) | 0.8016 | 0.9406 | 0.7298 | 0.9620 |
| | | | | (7.6250) | (12.6488) | (6.8578) | (10.0509) |
| | | | (0.5, 0.5, 0.5, 1, 1, 1) | 0.9300 | 0.9776 | 0.8918 | 0.9348 |
| | | | | (10.2607) | (16.3065) | (9.1466) | (13.9621) |
| | | | (1, 1, 1, 1, 1, 1) | 0.8328 | 0.9534 | 0.7564 | 0.9688 |
| | | | | (16.5917) | (26.4428) | (14.5405) | (24.5432) |
| | | | (1, 1, 1, 2, 2, 2) | 0.9314 | 0.9712 | 0.8906 | 0.9422 |
| | | | | (23.7564) | (36.4290) | (20.4332) | (36.4737) |
| | | | (2, 2, 2, 2, 2, 2) | 0.8450 | 0.9496 | 0.7682 | 0.9664 |
| | | | | (45.5142) | (70.4158) | (37.9084) | (76.2348) |
| | | (0.1, 0.1, 0.1, 0.3, 0.3, 0.3) | (0.5, 0.5, 0.5, 0.5, 0.5, 0.5) | 0.8306 | 0.9338 | 0.7700 | 0.9566 |
| | | | | (8.0197) | (12.3650) | (7.1669) | (11.2953) |
| | | | (0.5, 0.5, 0.5, 1, 1, 1) | 0.9064 | 0.9494 | 0.8652 | 0.9466 |
| | | | | (10.2536) | (14.6196) | (9.1805) | (14.3067) |
| | | | (1, 1, 1, 1, 1, 1) | 0.8454 | 0.9430 | 0.7788 | 0.9612 |
| | | | | (16.9729) | (25.0168) | (14.8551) | (28.0269) |
| | | | (1, 1, 1, 2, 2, 2) | 0.9172 | 0.9500 | 0.8754 | 0.9462 |
| | | | | (23.6133) | (31.8925) | (20.6296) | (38.2905) |
| | | | (2, 2, 2, 2, 2, 2) | 0.8698 | 0.9372 | 0.7942 | 0.9648 |
| | | | | (47.5882) | (66.9459) | (39.1228) | (95.2103) |
| | | (0.3, 0.3, 0.3, 0.3, 0.3, 0.3) | (0.5, 0.5, 0.5, 0.5, 0.5, 0.5) | 0.8292 | 0.8986 | 0.7616 | 0.9526 |
| | | | | (8.0671) | (11.1569) | (7.1500) | (10.8082) |
| | | | (0.5, 0.5, 0.5, 1, 1, 1) | 0.9262 | 0.9446 | 0.8912 | 0.9316 |
| | | | | (10.8288) | (14.3171) | (9.5091) | (15.1668) |
| | | | (1, 1, 1, 1, 1, 1) | 0.8450 | 0.9026 | 0.7812 | 0.9578 |
| | | | | (17.2150) | (22.4168) | (14.8647) | (26.5081) |
| | | | (1, 1, 1, 2, 2, 2) | 0.9316 | 0.9406 | 0.8936 | 0.9310 |
| | | | | (25.0334) | (30.8319) | (20.9536) | (40.4152) |
| | | | (2, 2, 2, 2, 2, 2) | 0.8696 | 0.9102 | 0.8010 | 0.9598 |
| | | | | (48.9337) | (58.5179) | (39.0391) | (86.8973) |
| | | (0.3, 0.3, 0.3, 0.5, 0.5, 0.5) | (0.5, 0.5, 0.5, 0.5, 0.5, 0.5) | 0.8694 | 0.9056 | 0.8156 | 0.9454 |
| | | | | (8.2082) | (10.7620) | (7.2262) | (10.8717) |
| | | | (0.5, 0.5, 0.5, 1, 1, 1) | 0.9082 | 0.9132 | 0.8642 | 0.9414 |
| | | | | (10.6453) | (12.9390) | (9.3519) | (14.2133) |
| | | | (1, 1, 1, 1, 1, 1) | 0.8786 | 0.9076 | 0.8226 | 0.9434 |
| | | | | (17.1771) | (21.1382) | (14.6899) | (25.9780) |
| | | | (1, 1, 1, 2, 2, 2) | 0.9248 | 0.9186 | 0.8838 | 0.9364 |
| | | | | (24.9900) | (27.5973) | (20.9123) | (38.3552) |
| | | | (2, 2, 2, 2, 2, 2) | 0.8968 | 0.9100 | 0.8360 | 0.9498 |
| | | | | (48.8431) | (53.4199) | (38.1265) | (84.5829) |

| Table 2 (continued) | | | | | | | |
|---|---|---|---|---|---|---|---|
| $(n_1, n_2, n_3, n_4, n_5, n_6)$ | $(\mu_1, \mu_2, \mu_3, \mu_4, \mu_5, \mu_6)$ | $(\delta'_1, \delta'_2, \delta'_3, \delta'_4, \delta'_5, \delta'_6)$ | $(\sigma^2_1, \sigma^2_2, \sigma^2_3, \sigma^2_4, \sigma^2_5, \sigma^2_6)$ | CP (AL) $CI_{FGCI}$ | $CI_{AM}$ | $CI_{BS1}$ | $CI_{BS2}$ |
| | | (0.5, 0.5, 0.5, 0.5, 0.5, 0.5) | (0.5, 0.5, 0.5, 0.5, 0.5, 0.5) | 0.8952 (8.2055) | 0.8970 (9.9677) | 0.8436 (7.1538) | 0.9408 (9.9022) |
| | | | (0.5, 0.5, 0.5, 1, 1, 1) | 0.9476 (11.0288) | 0.9296 (12.6493) | 0.9196 (9.4979) | 0.9152 (13.6809) |
| | | | (1, 1, 1, 1, 1, 1) | 0.8942 (16.9730) | 0.8856 (18.9134) | 0.8432 (14.3020) | 0.9466 (22.7060) |
| | | | (1, 1, 1, 2, 2, 2) | 0.9516 (25.2937) | 0.9354 (26.0932) | 0.9220 (20.7248) | 0.9248 (34.8531) |
| | | | (2, 2, 2, 2, 2, 2) | 0.9090 (48.6057) | 0.8974 (47.0657) | 0.8518 (36.9784) | 0.9478 (69.3067) |
| | | (0.1, 0.1, 0.3, 0.3, 0.5, 0.5) | (0.5, 0.5, 0.5, 0.5, 0.5, 0.5) | 0.8618 (8.0785) | 0.9338 (11.8067) | 0.8062 (7.1875) | 0.9398 (11.8587) |
| | | | (0.5, 0.5, 0.5, 1, 1, 1) | 0.9074 (10.4975) | 0.9450 (14.0510) | 0.8684 (9.3552) | 0.9362 (15.3534) |
| | | | (1, 1, 1, 1, 1, 1) | 0.8708 (17.0284) | 0.9336 (23.6226) | 0.8134 (14.7665) | 0.9498 (29.7163) |
| | | | (1, 1, 1, 2, 2, 2) | 0.9086 (23.8516) | 0.9372 (29.9712) | 0.8658 (20.6328) | 0.9400 (41.4774) |
| | | | (2, 2, 2, 2, 2, 2) | 0.8848 (47.6110) | 0.9292 (62.2276) | 0.8248 (38.6294) | 0.9496 (101.9620) |
| (30, 30, 30, 30, 30, 30) | (1, 1, 1, 1, 1, 1) | (0.1, 0.1, 0.1, 0.1, 0.1, 0.1) | (0.5, 0.5, 0.5, 0.5, 0.5, 0.5) | 0.8136 (4.8154) | 0.9999 (15.4392) | 0.7924 (4.6005) | 0.4276 (8.5477) |
| | | | (0.5, 0.5, 0.5, 1, 1, 1) | 0.9566 (6.1127) | 0.9999 (19.5638) | 0.9432 (5.8366) | 0.3178 (11.3839) |
| | | | (1, 1, 1, 1, 1, 1) | 0.7978 (10.1940) | 0.9998 (33.0285) | 0.7678 (9.7368) | 0.5198 (22.2396) |
| | | | (1, 1, 1, 2, 2, 2) | 0.9566 (13.7028) | 0.9999 (46.1774) | 0.9408 (13.0098) | 0.3356 (31.6191) |
| | | | (2, 2, 2, 2, 2, 2) | 0.8108 (25.8690) | 0.9999 (94.2322) | 0.7760 (24.5752) | 0.5912 (73.8401) |
| | | (0.1, 0.1, 0.1, 0.3, 0.3, 0.3) | (0.5, 0.5, 0.5, 0.5, 0.5, 0.5) | 0.8222 (4.8871) | 0.9999 (15.3067) | 0.7992 (4.6439) | 0.3026 (9.7968) |
| | | | (0.5, 0.5, 0.5, 1, 1, 1) | 0.9426 (6.0826) | 0.9999 (17.8290) | 0.9254 (5.7915) | 0.3602 (11.6771) |
| | | | (1, 1, 1, 1, 1, 1) | 0.8142 (10.1200) | 0.9999 (32.5466) | 0.7898 (9.6468) | 0.3886 (26.1704) |
| | | | (1, 1, 1, 2, 2, 2) | 0.9460 (13.4001) | 0.9999 (40.1522) | 0.9286 (12.7380) | 0.4108 (31.9072) |
| | | | (2, 2, 2, 2, 2, 2) | 0.8304 (25.0569) | 0.9999 (92.4834) | 0.7912 (23.7630) | 0.4350 (92.8136) |
| | | (0.3, 0.3, 0.3, 0.3, 0.3, 0.3) | (0.5, 0.5, 0.5, 0.5, 0.5, 0.5) | 0.8066 | 0.9998 | 0.7786 | 0.4798 |

(Continued)

| $(n_1, n_2, n_3, n_4, n_5, n_6)$ | $(\mu_1, \mu_2, \mu_3, \mu_4, \mu_5, \mu_6)$ | $(\delta'_1, \delta'_2, \delta'_3, \delta'_4, \delta'_5, \delta'_6)$ | $(\sigma^2_1, \sigma^2_2, \sigma^2_3, \sigma^2_4, \sigma^2_5, \sigma^2_6)$ | CP (AL) $CI_{FGCI}$ | $CI_{AM}$ | $CI_{BS1}$ | $CI_{BS2}$ |
|---|---|---|---|---|---|---|---|
| | | | | (4.8037) | (14.2032) | (4.5502) | (8.9447) |
| | | | (0.5, 0.5, 0.5, 1, 1, 1) | 0.9442 | 0.9999 | 0.9282 | 0.3554 |
| | | | | (6.0678) | (17.9521) | (5.7493) | (11.9905) |
| | | | (1, 1, 1, 1, 1, 1) | 0.7986 | 0.9999 | 0.7630 | 0.5626 |
| | | | | (9.7316) | (29.5119) | (9.2288) | (22.7197) |
| | | | (1, 1, 1, 2, 2, 2) | 0.9494 | 0.9998 | 0.9282 | 0.3774 |
| | | | | (13.0084) | (40.9760) | (12.2616) | (32.8189) |
| | | | (2, 2, 2, 2, 2, 2) | 0.8380 | 0.9998 | 0.7972 | 0.6042 |
| | | | | (23.8306) | (81.3217) | (22.3862) | (75.2510) |
| | | (0.3, 0.3, 0.3, 0.5, 0.5, 0.5) | (0.5, 0.5, 0.5, 0.5, 0.5, 0.5) | 0.8732 | 0.9999 | 0.8478 | 0.3694 |
| | | | | (4.8236) | (13.8424) | (4.5534) | (9.5585) |
| | | | (0.5, 0.5, 0.5, 1, 1, 1) | 0.9384 | 0.9996 | 0.9208 | 0.4376 |
| | | | | (6.0583) | (16.1432) | (5.7323) | (11.7522) |
| | | | (1, 1, 1, 1, 1, 1) | 0.8742 | 0.9999 | 0.8426 | 0.4332 |
| | | | | (9.6368) | (28.4826) | (9.1113) | (24.5225) |
| | | | (1, 1, 1, 2, 2, 2) | 0.9448 | 0.9998 | 0.9234 | 0.4502 |
| | | | | (12.9321) | (35.2431) | (12.2181) | (31.4213) |
| | | | (2, 2, 2, 2, 2, 2) | 0.8852 | 0.9999 | 0.8526 | 0.4724 |
| | | | | (23.0537) | (77.7036) | (21.6017) | (82.8702) |
| | | (0.5, 0.5, 0.5, 0.5, 0.5, 0.5) | (0.5, 0.5, 0.5, 0.5, 0.5, 0.5) | 0.8842 | 0.9999 | 0.8598 | 0.5324 |
| | | | | (4.7101) | (12.6219) | (4.4268) | (8.3001) |
| | | | (0.5, 0.5, 0.5, 1, 1, 1) | 0.9674 | 0.9999 | 0.9548 | 0.3984 |
| | | | | (5.9720) | (15.9091) | (5.6081) | (11.1150) |
| | | | (1, 1, 1, 1, 1, 1) | 0.8932 | 0.9998 | 0.8646 | 0.5998 |
| | | | | (9.2881) | (25.5004) | (8.7411) | (20.1494) |
| | | | (1, 1, 1, 2, 2, 2) | 0.9688 | 0.9996 | 0.9558 | 0.4210 |
| | | | | (12.4083) | (34.7170) | (11.5961) | (28.9112) |
| | | | (2, 2, 2, 2, 2, 2) | 0.9084 | 0.9998 | 0.8784 | 0.6178 |
| | | | | (21.7684) | (66.3863) | (20.2199) | (62.2931) |
| | | (0.1, 0.1, 0.3, 0.3, 0.5, 0.5) | (0.5, 0.5, 0.5, 0.5, 0.5, 0.5) | 0.8712 | 0.9999 | 0.8524 | 0.2040 |
| | | | | (4.9587) | (14.9867) | (4.7059) | (11.0931) |
| | | | (0.5, 0.5, 0.5, 1, 1, 1) | 0.9378 | 0.9999 | 0.9246 | 0.2912 |
| | | | | (6.1725) | (17.2994) | (5.8622) | (13.0781) |
| | | | (1, 1, 1, 1, 1, 1) | 0.8688 | 0.9999 | 0.8436 | 0.2546 |
| | | | | (10.0836) | (31.5761) | (9.6026) | (30.5581) |
| | | | (1, 1, 1, 2, 2, 2) | 0.9418 | 0.9999 | 0.9214 | 0.3374 |
| | | | | (13.4034) | (38.5120) | (12.7487) | (37.3294) |
| | | | (2, 2, 2, 2, 2, 2) | 0.8832 | 0.9999 | 0.8506 | 0.2834 |
| | | | | (24.5153) | (89.6123) | (23.2187) | (114.9333) |
| (50, 50, 50, 50, 50, 50) | (1, 1, 1, 1, 1, 1) | (0.1, 0.1, 0.1, 0.1, 0.1, 0.1) | (0.5, 0.5, 0.5, 0.5, 0.5, 0.5) | 0.8252 | 0.9999 | 0.8160 | 0.0760 |
| | | | | (3.8541) | (15.8903) | (3.7075) | (7.2803) |

| $(n_1, n_2, n_3, n_4, n_5, n_6)$ | $(\mu_1, \mu_2, \mu_3, \mu_4, \mu_5, \mu_6)$ | $(\delta'_1, \delta'_2, \delta'_3, \delta'_4, \delta'_5, \delta'_6)$ | $(\sigma^2_1, \sigma^2_2, \sigma^2_3, \sigma^2_4, \sigma^2_5, \sigma^2_6)$ | CP (AL) | | | |
|---|---|---|---|---|---|---|---|
| | | | | $CI_{FGCI}$ | $CI_{AM}$ | $CI_{BS1}$ | $CI_{BS2}$ |
| | | | (0.5, 0.5, 0.5, 1, 1, 1) | 0.9652 | 0.9999 | 0.9556 | 0.0696 |
| | | | | (4.9129) | (20.1920) | (4.7472) | (9.7267) |
| | | | (1, 1, 1, 1, 1, 1) | 0.8024 | 0.9999 | 0.7922 | 0.1286 |
| | | | | (8.3897) | (34.3204) | (8.1175) | (19.9112) |
| | | | (1, 1, 1, 2, 2, 2) | 0.9632 | 0.9999 | 0.9540 | 0.0850 |
| | | | | (10.9142) | (47.4026) | (10.5511) | (27.1045) |
| | | | (2, 2, 2, 2, 2, 2) | 0.8206 | 0.9999 | 0.7954 | 0.1518 |
| | | | | (21.6198) | (99.4624) | (20.9268) | (69.1200) |
| | | (0.1, 0.1, 0.1, 0.3, 0.3, 0.3) | (0.5, 0.5, 0.5, 0.5, 0.5, 0.5) | 0.8386 | 0.9999 | 0.8338 | 0.0434 |
| | | | | (3.9282) | (15.8730) | (3.7744) | (8.5329) |
| | | | (0.5, 0.5, 0.5, 1, 1, 1) | 0.9476 | 0.9999 | 0.9386 | 0.0852 |
| | | | | (4.8147) | (18.3732) | (4.6415) | (9.8208) |
| | | | (1, 1, 1, 1, 1, 1) | 0.8316 | 0.9999 | 0.8234 | 0.0648 |
| | | | | (8.3834) | (34.3113) | (8.1010) | (24.1798) |
| | | | (1, 1, 1, 2, 2, 2) | 0.9520 | 0.9999 | 0.9454 | 0.1002 |
| | | | | (10.7859) | (41.8454) | (10.4130) | (27.4038) |
| | | | (2, 2, 2, 2, 2, 2) | 0.8306 | 0.9999 | 0.8126 | 0.0860 |
| | | | | (21.0155) | (99.3153) | (20.3178) | (89.3190) |
| | | (0.3, 0.3, 0.3, 0.3, 0.3, 0.3) | (0.5, 0.5, 0.5, 0.5, 0.5, 0.5) | 0.8204 | 0.9999 | 0.8144 | 0.1090 |
| | | | | (3.8682) | (14.8042) | (3.7075) | (7.6852) |
| | | | (0.5, 0.5, 0.5, 1, 1, 1) | 0.9576 | 0.9999 | 0.9480 | 0.0946 |
| | | | | (4.8456) | (18.5610) | (4.6606) | (10.1607) |
| | | | (1, 1, 1, 1, 1, 1) | 0.8036 | 0.9999 | 0.7884 | 0.1670 |
| | | | | (8.0010) | (30.8982) | (7.7037) | (20.1601) |
| | | | (1, 1, 1, 2, 2, 2) | 0.9606 | 0.9999 | 0.9508 | 0.1088 |
| | | | | (10.4571) | (42.7048) | (10.0499) | (28.0026) |
| | | | (2, 2, 2, 2, 2, 2) | 0.8290 | 0.9999 | 0.8018 | 0.2020 |
| | | | | (19.6656) | (86.7010) | (18.9539) | (68.5778) |
| | | (0.3, 0.3, 0.3, 0.5, 0.5, 0.5) | (0.5, 0.5, 0.5, 0.5, 0.5, 0.5) | 0.8762 | 0.9999 | 0.8740 | 0.0820 |
| | | | | (3.8893) | (14.4091) | (3.7216) | (8.4659) |
| | | | (0.5, 0.5, 0.5, 1, 1, 1) | 0.9470 | 0.9999 | 0.9374 | 0.1240 |
| | | | | (4.7961) | (16.6455) | (4.6096) | (9.9949) |
| | | | (1, 1, 1, 1, 1, 1) | 0.8826 | 0.9999 | 0.8700 | 0.0966 |
| | | | | (7.9530) | (30.2705) | (7.6625) | (22.9065) |
| | | | (1, 1, 1, 2, 2, 2) | 0.9530 | 0.9999 | 0.9418 | 0.1488 |
| | | | | (10.3143) | (36.6303) | (9.9176) | (27.0717) |
| | | | (2, 2, 2, 2, 2, 2) | 0.8848 | 0.9999 | 0.8646 | 0.116 |
| | | | | (18.8334) | (83.0011) | (18.1105) | (79.5545) |
| | | (0.5, 0.5, 0.5, 0.5, 0.5, 0.5) | (0.5, 0.5, 0.5, 0.5, 0.5, 0.5) | 0.9070 | 0.9999 | 0.8962 | 0.1574 |
| | | | | (3.8144) | (13.1624) | (3.6426) | (7.2554) |
| | | | (0.5, 0.5, 0.5, 1, 1, 1) | 0.9786 | 0.9999 | 0.9720 | 0.1226 |

(Continued)

| Table 2 (continued) | | | | | | | |
|---|---|---|---|---|---|---|---|
| $(n_1, n_2, n_3, n_4, n_5, n_6)$ | $(\mu_1, \mu_2, \mu_3, \mu_4, \mu_5, \mu_6)$ | $(\delta'_1, \delta'_2, \delta'_3, \delta'_4, \delta'_5, \delta'_6)$ | $(\sigma^2_1, \sigma^2_2, \sigma^2_3, \sigma^2_4, \sigma^2_5, \sigma^2_6)$ | CP (AL) $CI_{FGCI}$ | $CI_{AM}$ | $CI_{BS1}$ | $CI_{BS2}$ |
| | | | | (4.7384) | (16.3421) | (4.5340) | (9.5195) |
| | | | (1, 1, 1, 1, 1, 1) | 0.8930 | 0.9999 | 0.8780 | 0.2186 |
| | | | | (7.5547) | (26.6011) | (7.2500) | (18.0184) |
| | | | (1, 1, 1, 2, 2, 2) | 0.9754 | 0.9999 | 0.9672 | 0.1400 |
| | | | | (9.8074) | (35.9742) | (9.3638) | (24.9016) |
| | | | (2, 2, 2, 2, 2, 2) | 0.9104 | 0.9999 | 0.8920 | 0.2498 |
| | | | | (17.6016) | (70.1342) | (16.8252) | (56.8175) |
| | | (0.1, 0.1, 0.3, 0.3, 0.5, 0.5) | (0.5, 0.5, 0.5, 0.5, 0.5, 0.5) | 0.8938 | 0.9999 | 0.8886 | 0.0200 |
| | | | | (4.0160) | (15.5456) | (3.8493) | (9.9117) |
| | | | (0.5, 0.5, 0.5, 1, 1, 1) | 0.9526 | 0.9999 | 0.9446 | 0.0608 |
| | | | | (4.8840) | (17.7752) | (4.6987) | (11.1697) |
| | | | (1, 1, 1, 1, 1, 1) | 0.8756 | 0.9999 | 0.8678 | 0.0316 |
| | | | | (8.3704) | (33.2824) | (8.0893) | (29.0916) |
| | | | (1, 1, 1, 2, 2, 2) | 0.9488 | 0.9999 | 0.9402 | 0.0792 |
| | | | | (10.7093) | (39.7266) | (10.3501) | (31.9445) |
| | | | (2, 2, 2, 2, 2, 2) | 0.8808 | 0.9999 | 0.8704 | 0.0376 |
| | | | | (20.3156) | (95.13447) | (19.6846) | (115.0963) |
| (100, 100, 100, 100, 100, 100) | (1, 1, 1, 1, 1, 1) | (0.1, 0.1, 0.1, 0.1, 0.1, 0.1) | (0.5, 0.5, 0.5, 0.5, 0.5, 0.5) | 0.8732 | 0.9999 | 0.8772 | 0.0012 |
| | | | | (2.7329) | (16.1904) | (2.6471) | (5.4062) |
| | | | (0.5, 0.5, 0.5, 1, 1, 1) | 0.9700 | 0.9999 | 0.9648 | 0.0008 |
| | | | | (3.5414) | (20.3978) | (3.4494) | (7.2442) |
| | | | (1, 1, 1, 1, 1, 1) | 0.8348 | 0.9999 | 0.8354 | 0.0030 |
| | | | | (6.2650) | (35.0941) | (6.0971) | (15.9937) |
| | | | (1, 1, 1, 2, 2, 2) | 0.9638 | 0.9999 | 0.9616 | 0.0020 |
| | | | | (8.0178) | (48.2604) | (7.8326) | (20.8600) |
| | | | (2, 2, 2, 2, 2, 2) | 0.8170 | 0.9999 | 0.8160 | 0.0022 |
| | | | | (16.7047) | (102.2348) | (16.3612) | (59.3653) |
| | | (0.1, 0.1, 0.1, 0.3, 0.3, 0.3) | (0.5, 0.5, 0.5, 0.5, 0.5, 0.5) | 0.8846 | 0.9999 | 0.8918 | 0.0000 |
| | | | | (2.8196) | (16.3093) | (2.7261) | (6.5520) |
| | | | (0.5, 0.5, 0.5, 1, 1, 1) | 0.9660 | 0.9999 | 0.9618 | 0.0010 |
| | | | | (3.4680) | (18.7847) | (3.3805) | (7.3270) |
| | | | (1, 1, 1, 1, 1, 1) | 0.8520 | 0.9999 | 0.8582 | 0.0010 |
| | | | | (6.2984) | (35.2816) | (6.1329) | (19.9033) |
| | | | (1, 1, 1, 2, 2, 2) | 0.9616 | 0.9999 | 0.9600 | 0.0028 |
| | | | | (7.9279) | (42.9625) | (7.7340) | (20.7974) |
| | | | (2, 2, 2, 2, 2, 2) | 0.8448 | 0.9999 | 0.8444 | 0.0010 |
| | | | | (16.3443) | (103.2218) | (16.0569) | (79.6386) |
| | | (0.3, 0.3, 0.3, 0.3, 0.3, 0.3) | (0.5, 0.5, 0.5, 0.5, 0.5, 0.5) | 0.8578 | 0.9999 | 0.8664 | 0.0026 |
| | | | | (2.7683) | (15.0435) | (2.6748) | (5.7390) |
| | | | (0.5, 0.5, 0.5, 1, 1, 1) | 0.9690 | 0.9999 | 0.9646 | 0.0024 |
| | | | | (3.5230) | (18.8232) | (3.4233) | (7.6004) |
| Table 2 (continued) | | | | | | | |
|---|---|---|---|---|---|---|---|
| $(n_1, n_2, n_3, n_4, n_5, n_6)$ | $(\mu_1, \mu_2, \mu_3, \mu_4, \mu_5, \mu_6)$ | $(\delta'_1, \delta'_2, \delta'_3, \delta'_4, \delta'_5, \delta'_6)$ | $(\sigma^2_1, \sigma^2_2, \sigma^2_3, \sigma^2_4, \sigma^2_5, \sigma^2_6)$ | CP (AL) $CI_{FGCI}$ | $CI_{AM}$ | $CI_{BS1}$ | $CI_{BS2}$ |
| | | | (1, 1, 1, 1, 1, 1) | 0.8214 | 0.9999 | 0.8286 | 0.0072 |
| | | | | (6.0201) | (31.7049) | (5.8552) | (16.1664) |
| | | | (1, 1, 1, 2, 2, 2) | 0.9644 | 0.9999 | 0.9596 | 0.0038 |
| | | | | (7.6498) | (43.1671) | (7.4550) | (21.2596) |
| | | | (2, 2, 2, 2, 2, 2) | 0.8132 | 0.9999 | 0.8148 | 0.0082 |
| | | | | (15.0719) | (88.6894) | (14.7550) | (57.1714) |
| | (0.3, 0.3, 0.3, 0.5, 0.5, 0.5) | (0.5, 0.5, 0.5, 0.5, 0.5, 0.5) | | 0.9180 | 0.9999 | 0.9234 | 0.0006 |
| | | | | (2.8449) | (14.8638) | (2.7437) | (6.7390) |
| | | | (0.5, 0.5, 0.5, 1, 1, 1) | 0.9618 | 0.9999 | 0.9598 | 0.0038 |
| | | | | (3.4498) | (16.9168) | (3.3505) | (7.5390) |
| | | | (1, 1, 1, 1, 1, 1) | 0.8982 | 0.9999 | 0.9022 | 0.0020 |
| | | | | (6.0485) | (31.1951) | (5.8747) | (19.4836) |
| | | | (1, 1, 1, 2, 2, 2) | 0.9620 | 0.9999 | 0.9538 | 0.0082 |
| | | | | (7.5600) | (37.2542) | (7.3573) | (20.7136) |
| | | | (2, 2, 2, 2, 2, 2) | 0.8984 | 0.9999 | 0.8926 | 0.0022 |
| | | | | (14.6875) | (86.9716) | (14.3815) | (73.0262) |
| | (0.5, 0.5, 0.5, 0.5, 0.5, 0.5) | (0.5, 0.5, 0.5, 0.5, 0.5, 0.5) | | 0.9250 | 0.9999 | 0.9244 | 0.0094 |
| | | | | (2.7656) | (13.3827) | (2.6605) | (5.6144) |
| | | | (0.5, 0.5, 0.5, 1, 1, 1) | 0.9828 | 0.9999 | 0.9804 | 0.0060 |
| | | | | (3.4635) | (16.5766) | (3.3548) | (7.3430) |
| | | | (1, 1, 1, 1, 1, 1) | 0.9070 | 0.9999 | 0.9082 | 0.0114 |
| | | | | (5.6762) | (27.1235) | (5.4975) | (14.7226) |
| | | | (1, 1, 1, 2, 2, 2) | 0.9806 | 0.9999 | 0.9746 | 0.0076 |
| | | | | (7.2151) | (36.3350) | (7.0047) | (19.4473) |
| | | | (2, 2, 2, 2, 2, 2) | 0.9242 | 0.9999 | 0.9154 | 0.0130 |
| | | | | (13.4330) | (71.8551) | (13.0897) | (48.3645) |
| | (0.1, 0.1, 0.3, 0.3, 0.5, 0.5) | (0.5, 0.5, 0.5, 0.5, 0.5, 0.5) | | 0.9394 | 0.9999 | 0.9410 | 0.0002 |
| | | | | (2.9524) | (16.0641) | (2.8508) | (8.0282) |
| | | | (0.5, 0.5, 0.5, 1, 1, 1) | 0.9658 | 0.9999 | 0.9632 | 0.0010 |
| | | | | (3.5098) | (18.1987) | (3.4133) | (8.4494) |
| | | | (1, 1, 1, 1, 1, 1) | 0.9094 | 0.9999 | 0.9152 | 0.0000 |
| | | | | (6.4386) | (34.4869) | (6.2743) | (25.4436) |
| | | | (1, 1, 1, 2, 2, 2) | 0.9642 | 0.9999 | 0.9590 | 0.0006 |
| | | | | (7.9464) | (41.02513) | (7.7511) | (24.8286) |
| | | | (2, 2, 2, 2, 2, 2) | 0.9058 | 0.9999 | 0.9016 | 0.0000 |
| | | | | (16.2537) | (100.0387) | (15.9633) | (111.3212) |

For $k = 6$, the performances of $CI_{AM}$ were better than those of the others for small sample sizes, but its coverage probabilities were equal to 1 for large sample sizes. Meanwhile, those of $CI_{FGCI}$ and $CI_{BS1}$ were greater than the nominal confidence level of 0.95 for large sample sizes but the average lengths of $CI_{BS1}$ were shorter than those of

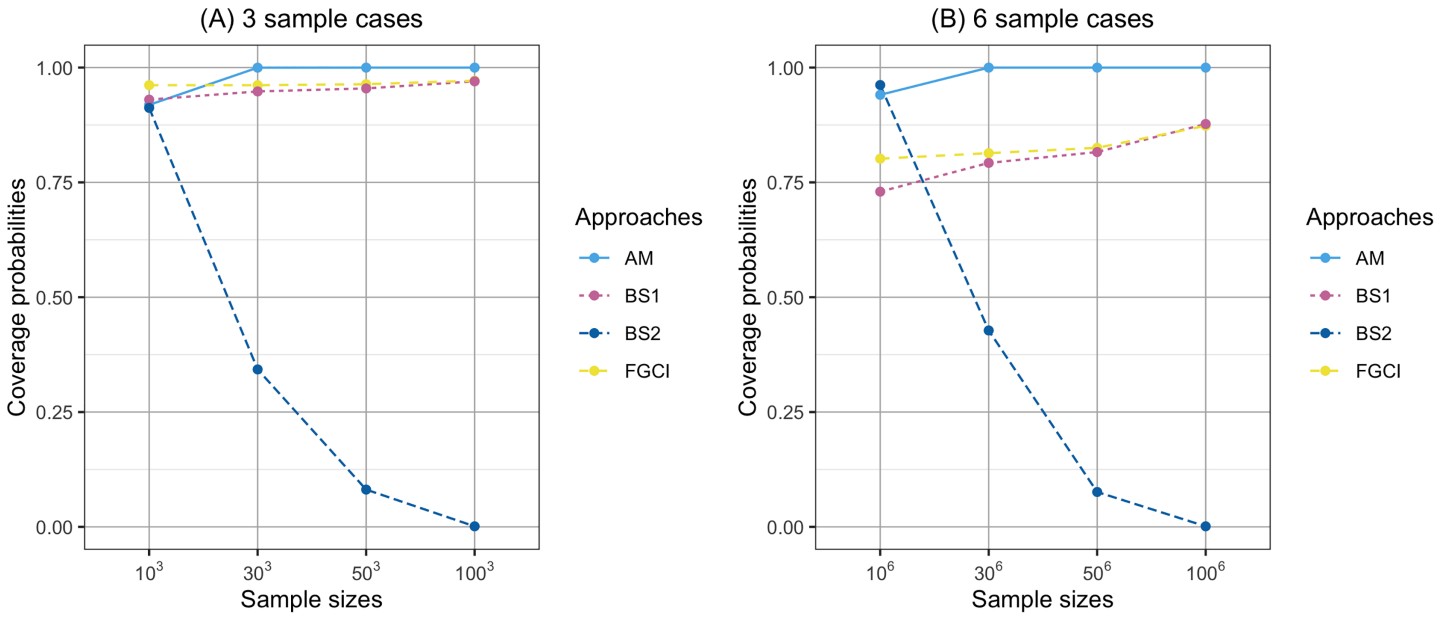

**Figure 1 Comparison of the coverage probabilities of proposed approaches according to sample sizes.** (A) Three sample cases, (B) six sample cases.

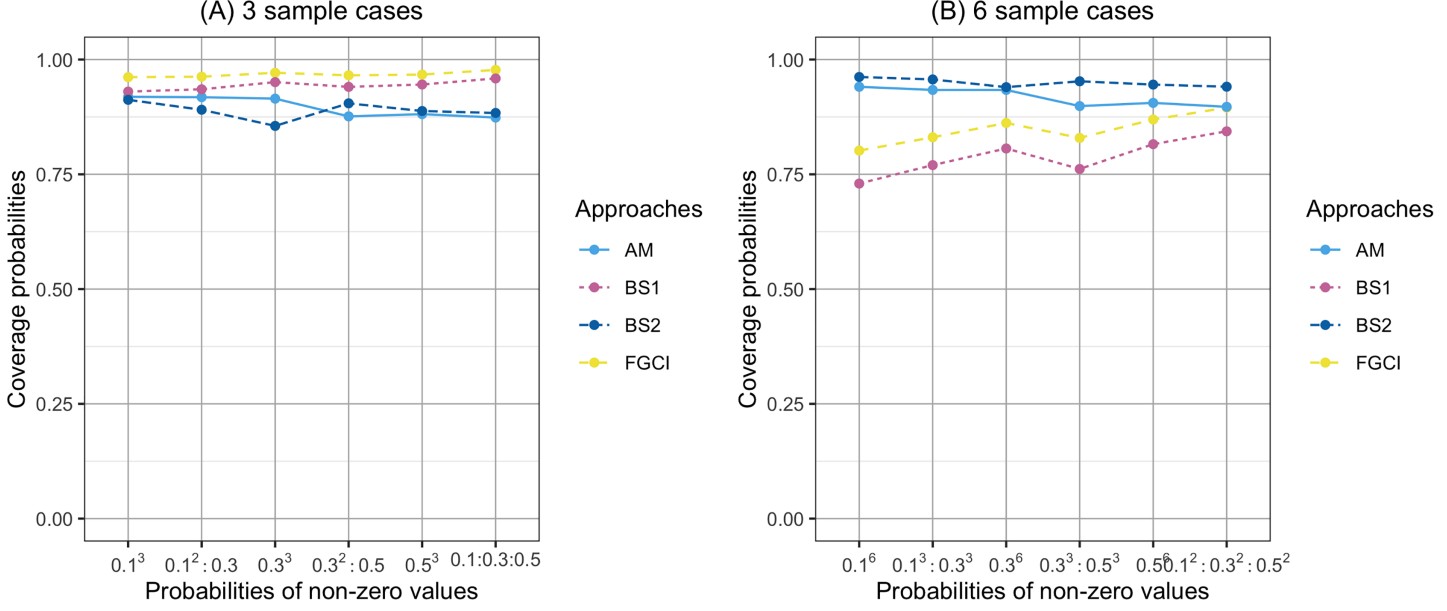

**Figure 2 Comparison of the coverage probabilities of proposed approaches according to probabilities of non-zerovalues.** (A) Three sample cases, (B) six sample cases.

$CI_{FGCI}$. Moreover, the coverage probabilities of $CI_{BS2}$ were less than the nominal confidence level of 0.95.

In Fig. 1, the coverage probabilities of $CI_{BS2}$ were smaller when the sample sizes are larger because its average lengths were shorter which was effect to the proportion of the time that the interval contains the true value. In addition, the prior distribution was effect

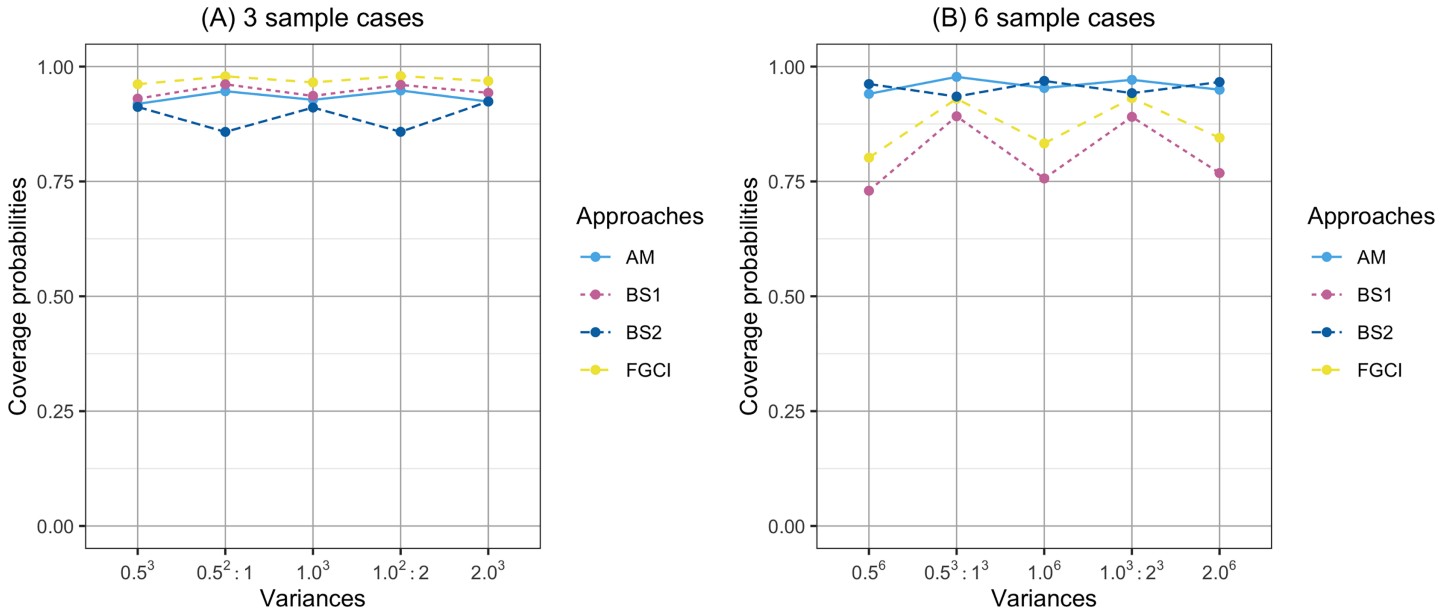

**Figure 3 Comparison of the coverage probabilities of proposed approaches according to variance.** (A) Three samples cases, (B) six sample cases.

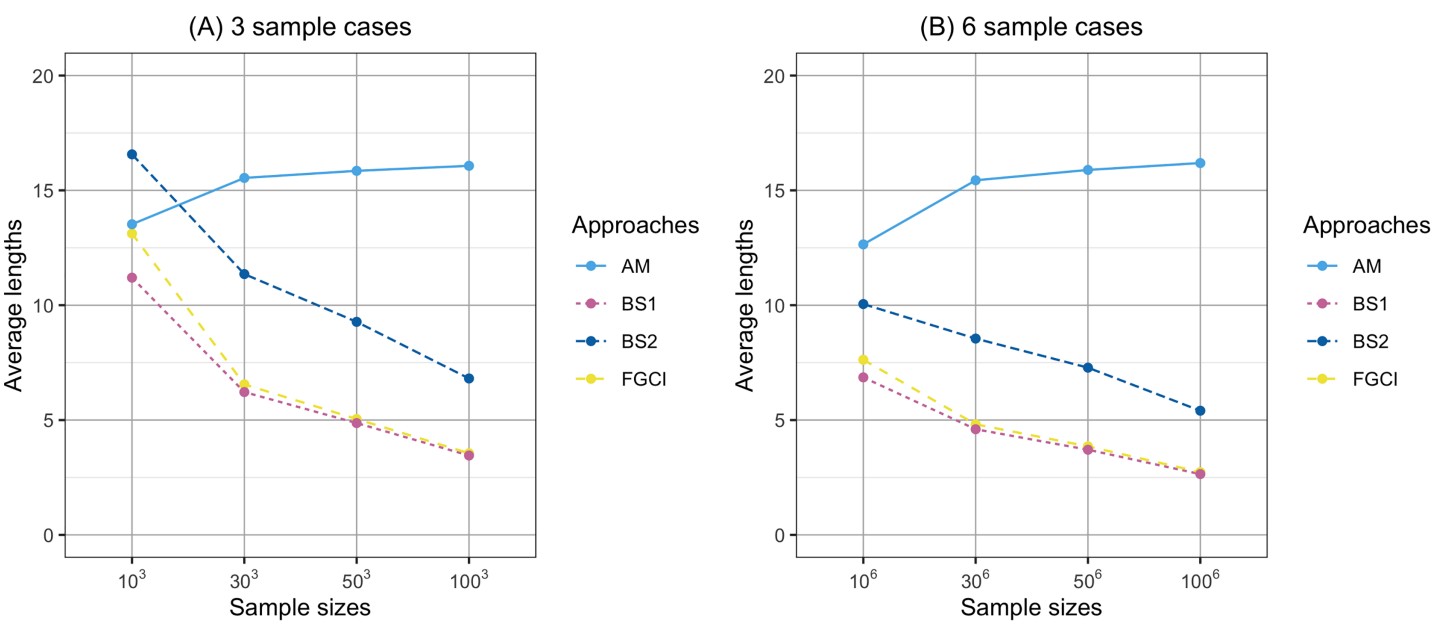

**Figure 4 Comparison of the average lengths of proposed approaches according to sample sizes.** (A) Three samples cases, (B) six samples cases.

to the coverage probabilities; see *Yosboonruang, Niwitpong & Niwitpong (2020)*. Moreover, in Fig. 4, the average lengths of $CI_{AM}$ were wider when the sample sizes are larger because the confidence interval uses the concepts of the central limit theorem and is based on the initial confidence interval for the single parameter. The initial confidence interval was effect to the coverage probabilities and the average lengths; see *Thangjai, Niwitpong & Niwitpong (2020b)*.

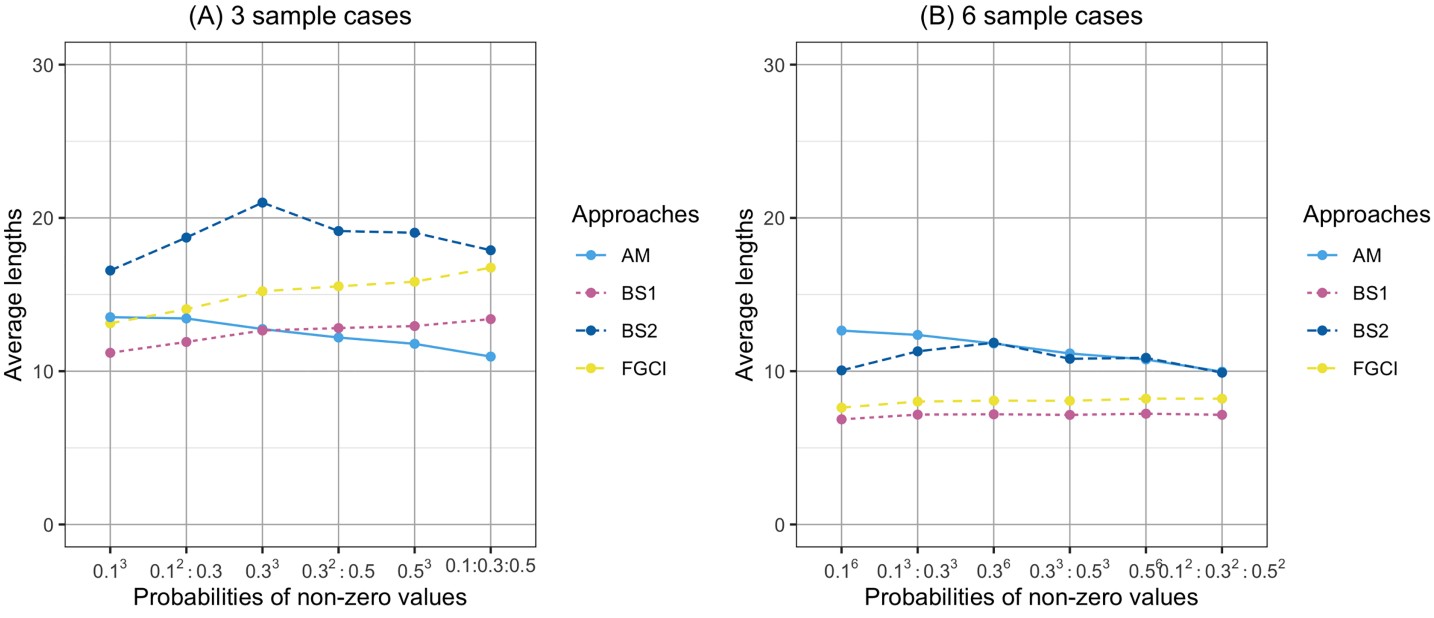

**Figure 5 Comparison of the average lengths of proposed approaches according to probabilities of non-zero values.** (A) Three sample cases, (B) six sample cases.

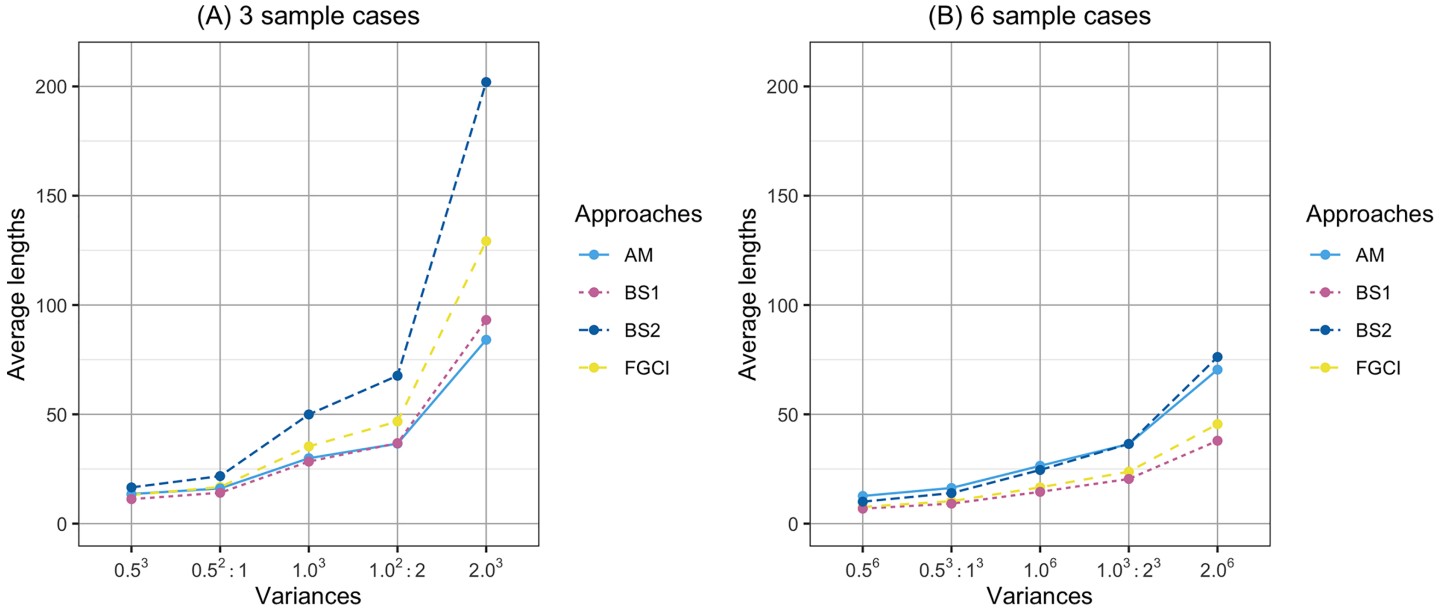

**Figure 6 Comparison of the average lengths of proposed approaches according to variance.** (A) Three sample cases. (B) six sample cases.

## Empirical application

The efficacies of the proposed confidence intervals are illustrated using real datasets. Beforehand, a Monte Carlo simulation consisting of 1,000 repetitions was run to compute the FGCI and Bayesian confidence intervals.

**Table 3 The rainfall data of five regions (mm).**

| Northern | | Northeastern | | Central | | Eastern | | Southern | |
|---|---|---|---|---|---|---|---|---|---|
| 17.5 | 0.0 | 1.2 | 62.8 | 2.0 | 4.4 | 16.4 | 0.3 | 0.1 | 8.4 |
| 6.1 | 0.0 | 0.2 | 0.0 | 9.3 | 31.2 | 1.3 | 12.9 | 4.2 | 0.5 |
| 1.6 | 1.2 | 0.0 | 4.4 | 0.0 | 8.6 | 29.1 | 12.6 | 19.7 | 8.4 |
| 0.0 | 5.6 | 25.1 | 0.0 | 6.0 | 11.7 | 11.4 | 0.0 | 0.7 | 31.3 |
| 0.2 | 0.4 | 39.0 | 4.5 | 3.0 | 1.2 | 118.3 | 6.7 | 0.0 | 0.3 |
| 3.6 | 0.4 | 0.8 | 0.5 | 9.6 | 1.4 | 7.8 | 4.2 | 0.3 | 1.6 |
| 0.2 | 0.3 | 0.0 | 19.6 | 21.4 | 8.7 | 1.0 | 0.2 | 1.5 | 8.1 |
| 0.0 | 3.5 | 6.8 | 0.6 | 1.2 | 2.2 | 1.3 | | 0.0 | 1.0 |
| 0.8 | 0.0 | 1.2 | 27.6 | 35.9 | 11.3 | | | 0.0 | 5.5 |
| 0.3 | 2.5 | 0.6 | 0.4 | 7.6 | 8.8 | | | 0.0 | 0.0 |
| 7.0 | 5.6 | 7.2 | 0.0 | 1.8 | 0.4 | | | 0.0 | 145.2 |
| 0.8 | 0.2 | 0.0 | 0.0 | | | | | 65.8 | 23.2 |
| 3.3 | 4.1 | 0.0 | 0.0 | | | | | 2.4 | 7.6 |
| 18.4 | 15.6 | 0.8 | 0.0 | | | | | 24.6 | 33.8 |
| 0.0 | | | | | | | | 12.0 | |

Note:
   Source, Thai Meteorological Department (https://www.tmd.go.th/climate/climate.php).

**Table 4 Sample statistics of five regions.**

| Statistics | Northern | Northeastern | Central | Eastern | Southern |
|---|---|---|---|---|---|
| $n_i$ | 29 | 28 | 22 | 15 | 29 |
| $n_{i(1)}$ | 23 | 18 | 21 | 14 | 16 |
| $n_{i(0)}$ | 6 | 10 | 1 | 1 | 13 |
| $\bar{y}_{-i(1)}$ | 0.56 | 1.10 | 1.63 | 1.54 | 2.07 |
| $s^2_{i(1)}$ | 2.28 | 3.26 | 1.38 | 3.21 | 3.66 |

The datasets comprise rainfall data on 1 September 2021 for five regions in Thailand reported by the Thai Meteorological Department. The data are reported in Table 3 and the statistics in Table 4. The densities of these datasets are presented in Fig. 7. First, the possible distributions for the positive rainfall data were considered by applying the minimum Akaike information criterion (AIC), the results for which are reported in Table 5. It can be seen that the log-normal distribution fits the positive rainfall data for the northern, northeastern, and eastern regions of Thailand. Moreover, histograms of the rainfall data series and QQ-plots of the log-transformed rainfall data series in Figs. 8 and 9, respectively, follow normal distributions. Next, the zero values in the rainfall datasets fit binomial distributions, and so overall, the rainfall datasets follow delta-lognormal distributions.

The 95% two-sided confidence intervals for the common percentile of delta-lognormal distributions based on the four methods were $CI_{FGCI} = [5.7849, 86.7576]$ with an interval length of 80.9727, $CI_{AM} = [0.0347, 161.3430]$ with an interval length of 161.3083,

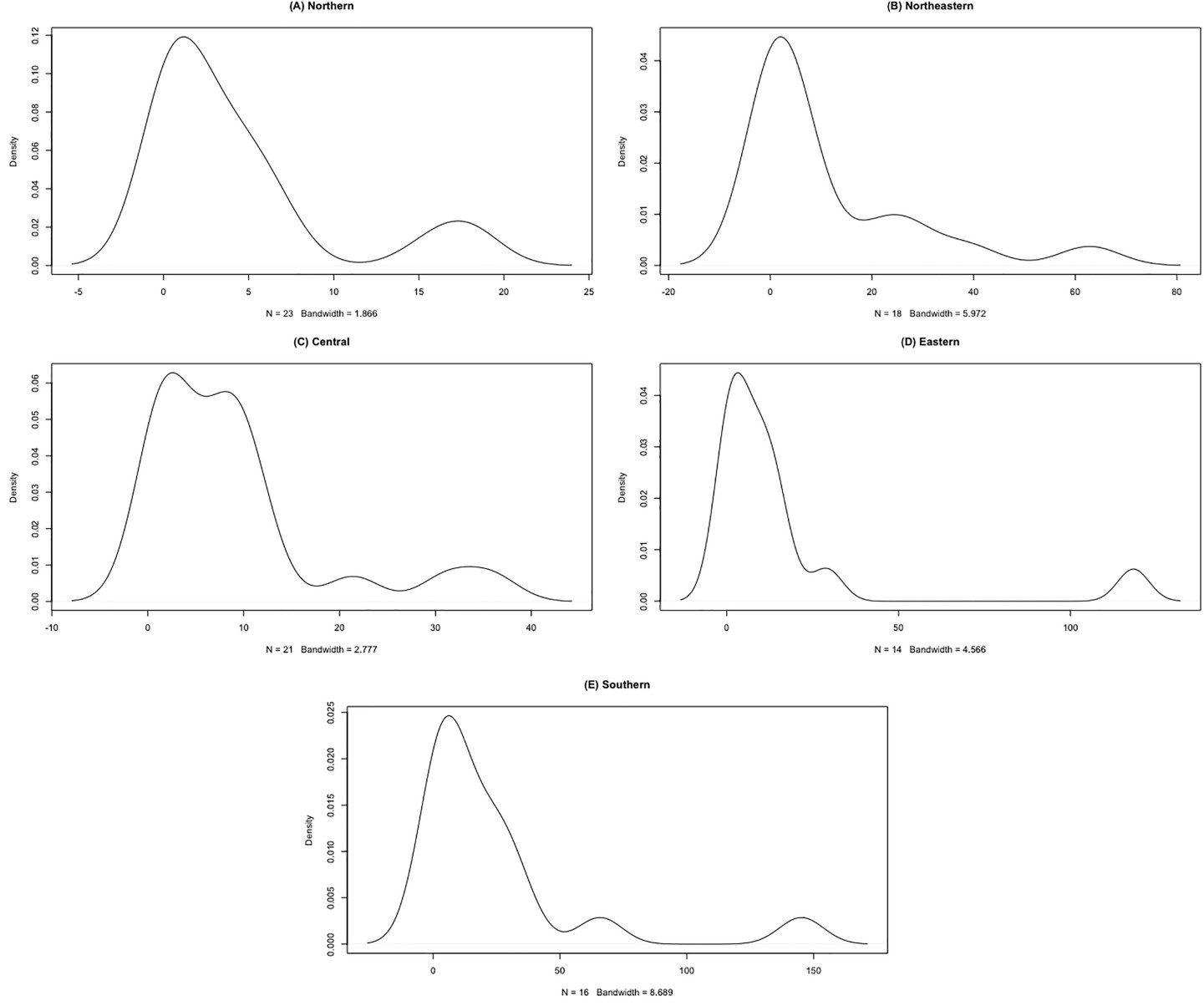

**Figure 7 Densities of rainfall data of five regions.** (A) Northern (B) Northeastern (C) Central (D) Eastern (E) Southern.

$CI_{BS1} = [5.3282, 66.9045]$ with an interval length of 61.5763, and $CI_{BS2} = [23.5146, 397.8828]$ with an interval length of 374.3682. Thus, the results of the proposed approaches to construct the confidence intervals are in accordance with those from the simulation study for a small sample size. The Bayesian confidence interval based on fiducial quantity had the shortest average length.

In this article, we are interested in the 95% confidence interval for the common percentile of rainfall data for the $p = 0.95$ percentile. The common percentile of rainfall data is weighted average of the percentiles for rainfall data of five regions. Generally, the $p$-th percentile is any number between 0 and 100. For $p = 0.95$, the $p$-th percentile is in

**Table 5 The AIC values of five regions.**

| Distribution | Northern | Northeastern | Central | Eastern | Southern |
|---|---|---|---|---|---|
| Normal | 146.9695 | 156.6773 | 157.6655 | 138.4266 | 163.5237 |
| Log-normal | 113.0194 | 115.0962 | 137.7288 | 102.1466 | 135.4856 |
| Gamma | 114.4763 | 118.5788 | 137.9854 | 104.0756 | 133.4562 |
| Exponential | 114.2356 | 124.2752 | 134.9935 | 106.5699 | 135.0211 |

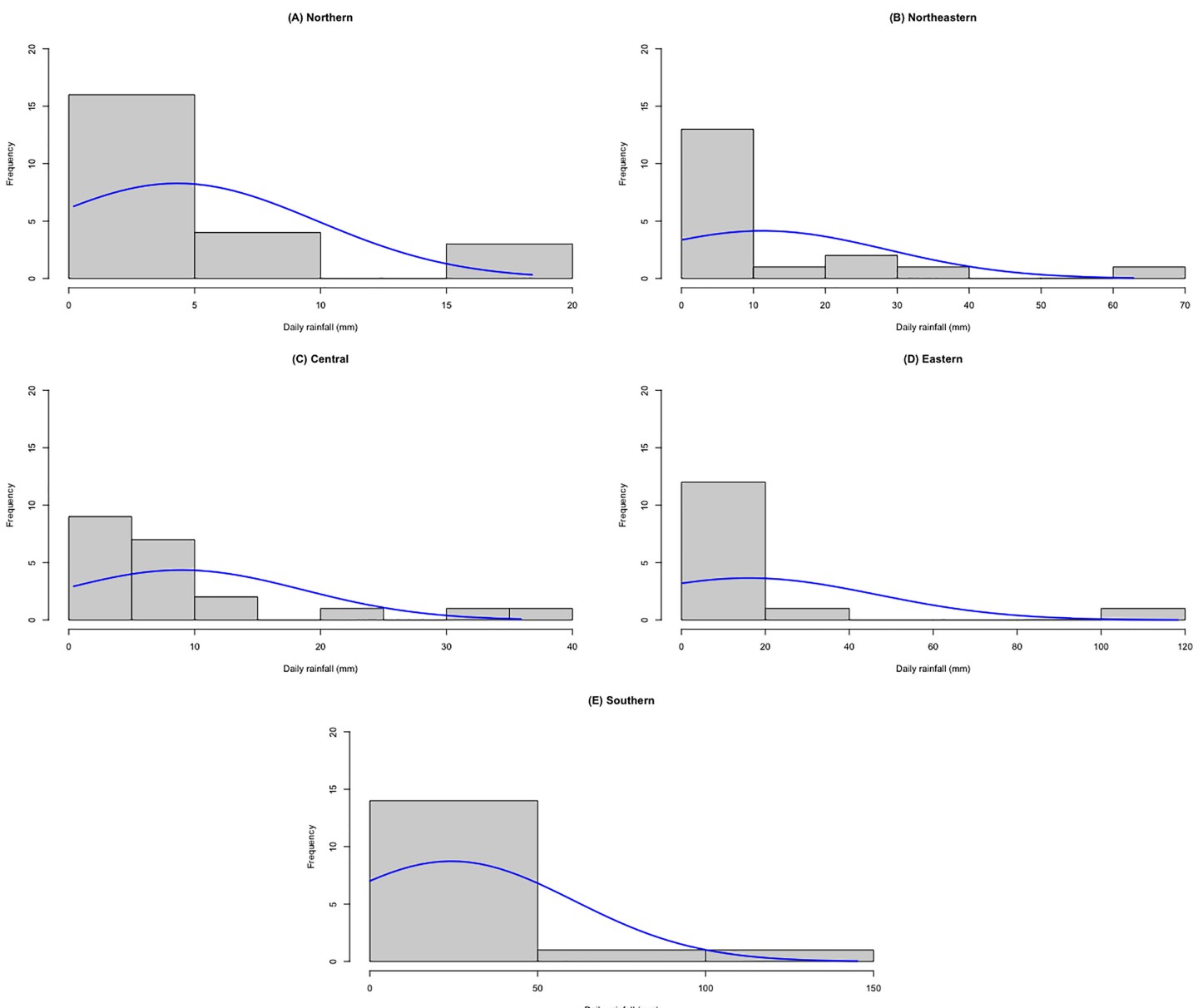

**Figure 8 Histograms of rainfall data of five regions.** (A) Northern (B) Northeastern (C) Central (D) Eastern (E) Southern.

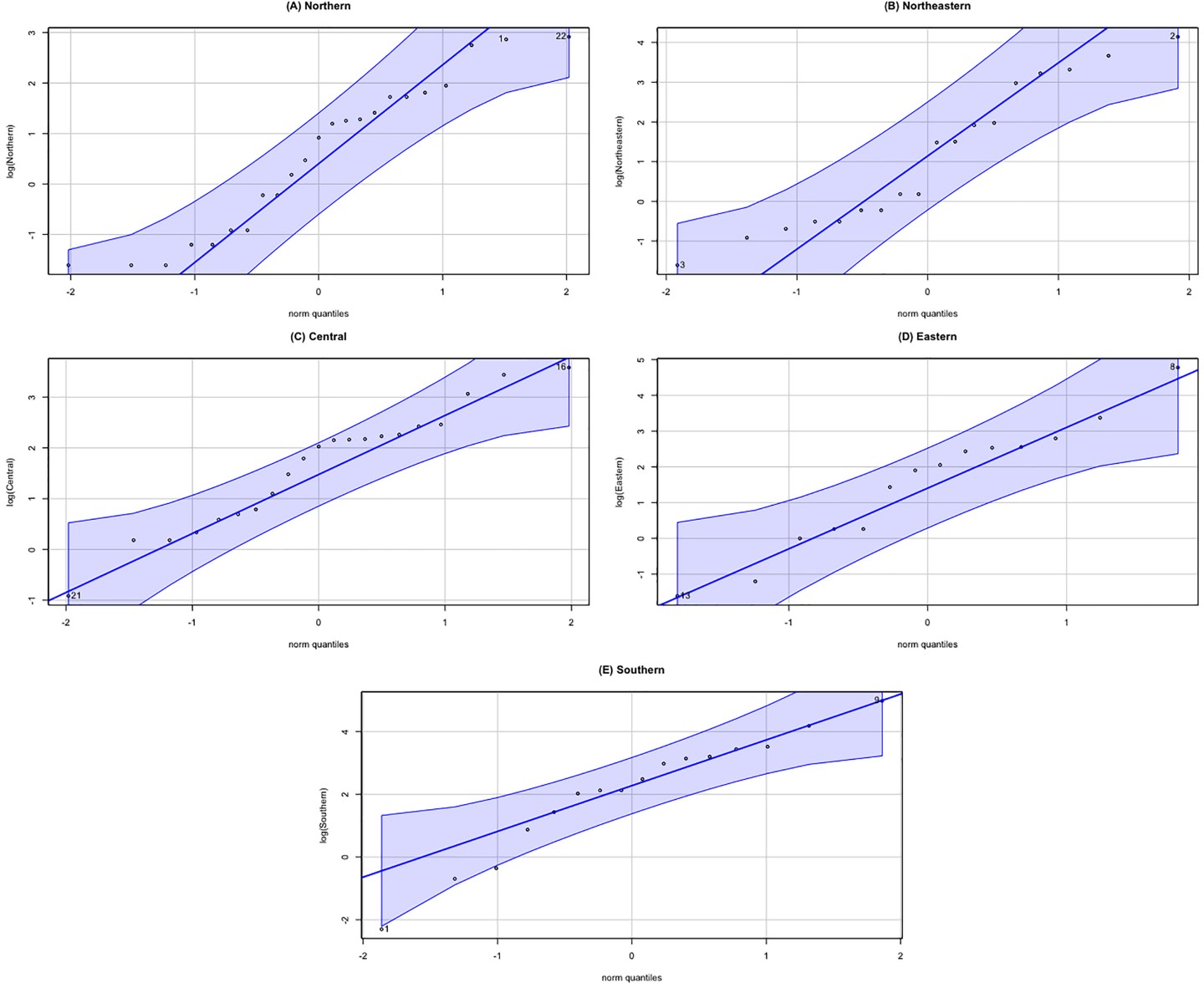

**Figure 9 Normal QQ-plots of log-transformation rainfall data of five regions.** (A) Northern (B) Northeastern (C) Central (D) Eastern (E) Southern.

range between 2.50 and 97.50. Therefore, the lower and upper limits of the confidence interval are the 2.50-th and 97.50-th percentiles of the common percentile for rainfall data. It can be confirmed that the lower and upper limits of the common percentile for rainfall data will work. The confidence interval therefore is in range between the lower limit and the upper limit. In interpreting practice, it was supposed to have at least a 95% chance of covering the 95th percentile. If the percentile is less than the lower limit, then we will have the observed 2.50 or fewer values in rainfall data that are below the 95th percentile. If the percentile exceeds the upper limit, then we wilhave the observed 97.50 or more out of 100 values in rainfall data that are below the 95th percentile.

## DISCUSSION

This paper proposed the two Bayesian approaches using the fiducial quantity and the approximate fiducial distribution. The prior distributions for $\lambda_{p_i}$ of two Bayesian approaches are difference. Then, the performances of the two Bayesian approaches are difference.

As a limitation of this paper, the FGCI approach can be used to construct the confidence interval for complex parameters, but the FGCI approach is based on the simulation techniques. Moreover, the numerical simulation is based on the maximum likelihood estimate only. The adjusted MOVER approach is easy to use the exact formula for constructing the confidence interval, but the adjusted MOVER approach requires initial confidence interval of single parameter. The Bayesian approach can be used to estimate the confidence interval for complex parameters, but the Bayesian approach is based on the simulation techniques. Moreover, the numerical simulation is based on the prior distribution.

For the sake of saving space, the simulation results for $k = 4$, $k = 5$, and $k > 6$ are not displayed. The results for $k = 4$ and $k = 5$ are similarly the results for $k = 3$. As the sample case increased, the results for $k > 6$ are similarly the results for $k = 6$.

Considering sample sizes, the performance of the proposed approaches when the sample size in each group is different are similarly the performance of the proposed approaches when the sample size in each group is equal.

The FGCI and Bayesian approaches were derived using a simulation technique whereas the adjusted MOVER approach was derived using formulas. The results in this investigation are similar to those of *Harvey & van der Merwe (2012)*; *Rao & D'Cunha (2016)*, *Thangjai & Niwitpong (2020b)*, *Thangjai, Niwitpong & Niwitpong (2021a)* and *Thangjai, Niwitpong & Niwitpong (2021b)*.

## CONCLUSION

We provided confidence intervals for the common percentile of several delta-lognormal distributions using FGCI, adjusted MOVER, and two Bayesian approaches. The results indicate that $CI_{BS1}$ was better than the others for $k = 3$ or $k = 6$. Overall, the Bayesian approach was superior to the classical approach for constructing confidence intervals for the common percentile of several delta-lognormal distributions. In future research, we will consider the statistical inference for percentiles of delta-lognormal distribution and other distributions.

### Funding

This research was funded by King Mongkut's University of Technology North Bangkok. Grant No, KMUTNB-66-KNOW-01. The funders had no role in study design, data collection and analysis, decision to publish, or preparation of the manuscript.

## Grant Disclosures

The following grant information was disclosed by the authors:

King Mongkut's University of Technology North Bangkok: KMUTNB-66-KNOW-01.

## Competing Interests

The authors declare that they have no competing interests.

## Author Contributions

- Warisa Thangjai conceived and designed the experiments, performed the experiments, analyzed the data, prepared figures and/or tables, authored or reviewed drafts of the article, and approved the final draft.
- Sa-Aat Niwitpong conceived and designed the experiments, analyzed the data, authored or reviewed drafts of the article, and approved the final draft.
- Suparat Niwitpong performed the experiments, analyzed the data, prepared figures and/or tables, authored or reviewed drafts of the article, and approved the final draft.

## Data Availability

The data is available in the Supplemental File.

## Supplemental Information

Supplemental information for this article can be found online at http://dx.doi.org/10.7717/peerj.14498#supplemental-information.

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
