# Peer review of "Estimation of common percentile of rainfall datasets in Thailand using delta-lognormal distributions"

_PeerJ, doi:10.7717/peerj.14498_

## Round 0.1 · original submission · Major Revisions

The work is interesting, has a clear degree of originality, and is appropriate for publication in the journal after performing a major and very careful revision.

Nevertheless, it needs some further improvements. In general, there are still some occasional grammar errors throughout the manuscript, especially the article "the," "a," and "an" are missing in many places; please make spellchecking in addition to these minor issues. The reviewer has listed some specific comments that might help the authors further enhance the manuscript's quality.

1. Specific Comments

• Overall, the Abstract section is not giving any information about methodology, results, conclusion, and recommendations as it should be clear. I suggest the authors remove generic lines and present the strong statements and novelty of the article. The abstract is written in qualitative sentences. It needs to be modified and rewritten based on the most important quantitative results from this research. The abstract should be redesigned. You should avoid using acronyms in the abstract and insert the work's main conclusion.

• You have used many abbreviations in the text. From this perspective, an Index of Notations and Abbreviations would be beneficial for a better understanding of the proposed work. Furthermore, please check carefully if all the abbreviations and notations considered in the work are explained for the first time when they are used, even if these are considered trivial by the authors. The paper should be accessible to a wide audience. Furthermore, it will make sense to include also the notations in this index.

• The objectives should be more explicitly stated.

• The Introduction section must be written in a more quality way. The research gap should be delivered in a more clear way with the necessity for the conducted research work.

• What is the novelty of this work?

• It is better to improve your contributions which are not so clear to show the advantage of your work.

• The novelty of this work must be clearly addressed and discussed in the Introduction section.

• The methodology limitation should be mentioned.

Many equations are presented in the paper, and most look OK. However, please check carefully whether all equations are necessary and whether the quantities involved are properly explained. Also, some equations need references.

• Discussion

• Overall, the discussion part is weak. The Discussion should summarize the manuscript's main finding(s) in the context of the broader scientific literature and address any study limitations or results that conflict with other published work.

Reviewer 1 ·

Basic reporting

This paper suggests the confidence intervals (CIs) for the common percentile in the delta-lognormal distributions. The related methods for constructing the CIs are FGCI, Adj-MOVER, and two Bayesian approaches. Then the authors applied these CIs to the rainfall data. The methods used in this paper are not new, but the authors present the new estimators. I have no concern to the methods used in this paper, as they are general. But the useful and limitation of the estimators proposed in this paper are still needed to clarify. The lists are given below.
It seems the authors are interested to study the rainfall data occurred in Thailand. This is related to the sentence given in the beginning of the first section and the application section. Yes, water (from the rain) is necessary to study as it is useful in agriculture and can damage (flooding), for example. The rainfall is then usually reported the values in statistics, such as minimum, maximum, mean, standard deviation, or comparing the spared of the data by coefficient of variation. Very important to note here that these statistics have the well-meaning in conclusion the character of the rainfall data. Now, my main question is that what is the meaning of the percentile of the rainfall data in the author’s view? I try to find where the authors describe this in the paper. Unfortunately, I see only a very small paragraph at the end of the introduction section. In data analysis, the authors provide only the CI values from the four methods but did not give any expansion. In fact, the latter section could be the highlight of this work as it is related the title. So, it is needed to be clear. The motivation and example in applications must be added in the introduction, especially who used or estimated the percentile of the rainfall data, or other pollutions. Again in the analysis, could the authors explain the meaning of the 95% confidence interval for the common percentile for rainfall; CI = [5.78, 86.76] and CI = [23.51, 397.88]? Is this the r-th percentile in interval? If yes, what does it means if r > 100? Generally, r% is in [0, 100]% If no, what is r-th percentile of these datasets and why the upper limit is greater than the maximum value of the data. How to manage the value of percentile that greater than the maximum in simulation work?
See for example:
> x = c(16.4,1.3,29.1,11.4,118.3,7.8,1,1.3,0.3,12.9,12.6,0,6.7,4.2,0.2)
> quantile(x, probs = c(0,0.25,0.50,0.75,1))
0% 25% 50% 75% 100%
0.00 1.15 6.70 12.75 118.30
> quantile(x, probs = c(300))
Error in quantile.default(x, probs = c(300)) : 'probs' outside [0,1]

What is a very small Discussion and Conclusion sections? It is too short. I suggest the authors give more discussion which are different from the simulation results. We need to know that why only one Bayesian method is good, meanwhile another one Bayesian is bad performance. The authors have to see why it close to zero in some situations. What is the limitation of this work? When the populations are 4, 5 and greater than 6 which method should be used, why? Furthermore, when the sample size in each group is different, performance of estimator may be down. Is it happening with this work? The authors will also see in your datasets the sample size are quite different. I hope these questions will be useful to the user who need to apply in applications.

Experimental design

See above.

Validity of the findings

See above.

Additional comments

See above.

Reviewer 2 ·

Basic reporting

The paper have a good methodology and professional English.

Experimental design

The experimental design is suitable for simulation study.

Validity of the findings

The statistical method is correct.

Additional comments

1. The paper title should have the word “delta-lognormal distribution”.
2. In line 63, the symbol ‘k’ is not defined.
3. In Figure 1, why the coverage probabilities of BS2 are smaller when the sample sizes are larger? The author should explain the reason.
4. In Figure 4, why the average widths of AM are wider when the sample sizes are larger? The author should explain the reason.

---

## Round 0.2 · Minor Revisions

Please check the reviewer's comments carefully.

Reviewer 1 ·

Basic reporting

-

Experimental design

-

Validity of the findings

-

Additional comments

The application is still unclear for me. Especially, the meaning of r-th for the rainfall data.

---

## Round 0.3 · accepted · Accept

I congratulate the authors for the effort put into this paper! The manuscript is significantly improved; therefore, I recommend accepting it in its current form!